# PADA-Coder: Improving Plan-Following Code Generation via Perturbation-Verified Attention Distillation and Dynamic Alignment

**Yihong Huang** [1]   **Ke Qin** [1 2]   **Rongzheng Wang** [1]   **Muquan Li** [1]   **Jiakai Li** [1]   **Xiurui Xie** [1 2]   **Shuang Liang** [1 2]

## Abstract

The Plan-then-Code paradigm effectively enhances Large Language Models (LLMs) in complex code generation by decomposing reasoning into explicit, interpretable steps. However, introducing the plan and verification report substantially enlarges the context, which in turn misdirects the model's attention toward irrelevant tokens and the most recently generated code. This effect leads the model to overlook critical constraints and to generate incorrect code, especially for small-scale LLMs (less than 8B). To address this issue, we propose **P**erturbation-Verified **A**ttention **D**istillation and Dynamic **A**lignment (PADA). PADA identifies the key tokens most critical to the student model and constructs the optimal attention target matrix, dynamically aligning the student's focus with key tokens for each plan step. We evaluate PADA with two teacher models and three student models across seven benchmarks, and the results show that PADA improves Pass@1 by up to 16.7% and outperforms SOTA methods in overall average performance.

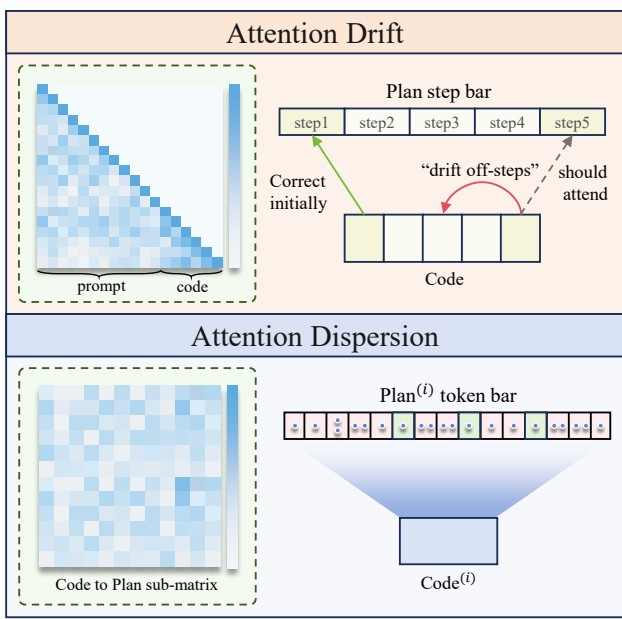

*Figure 1.* Observation of Attention Allocation Imbalance. Top (Attention Drift): As sequence length grows, model increasingly attends to the self-generated history (dense lower-triangular region). Bottom (Attention Dispersion): Model distributes attention relatively uniformly, resulting in insufficient focus on key tokens.

## 1. Introduction

Complex code-generation tasks performed by large language models (LLMs) require robust modeling of long-range dependencies and rigorous compliance with formal logical specifications. To improve the performance of LLMs on complex code generation tasks, the "Plan-then-Code" paradigm (Wang et al., 2023; Lei et al., 2025) has been widely adopted. In this framework, models first formulate a natural language plan to explicitly decompose the reasoning process into interpretable steps, subsequently implementing

the code based on this planned trajectory.

Nevertheless, introducing explicit plans creates a new challenge: LLMs frequently fail to maintain strict adherence to the plan during code generation. Specifically, the model's attention is an inherently limited resource. Plan-then-Code lengthens the context, forcing a finite attention budget to spread over more tokens. This often leads to **Attention Allocation Imbalance**, especially for small-scale LLMs (less than 8B). Recent research has sought to address this dilemma in small-scale LLMs from the perspective of attention mechanisms, primarily through two classes of techniques: attention guidance and attention distillation. SPA (Tian & Zhang, 2025) simulates attention guidance through logarithmic arithmetic, SELF-ANCHOR(Zhang et al., 2025) aligns the model's attention to the plan, and LeaF(Guo et al., 2025) introduces causal verification for attention distillation.

However, existing approaches typically address attention

[1]University of Electronic Science and Technology of China [2]Ubiquitous Intelligence and Trusted Services Key Laboratory of Sichuan Province. Correspondence to: Shuang Liang <shuangliang@uestc.edu.cn>.

*Proceedings of the 43rd International Conference on Machine Learning*, Seoul, South Korea. PMLR 306, 2026. Copyright 2026 by the author(s).

constraints partially, failing to provide a cohesive solution. as shown in Figure 1, to effectively address the problem of Attention Allocation Imbalance, it is necessary to consider two issues. The first is **Attention Dispersion**: Attention is distributed broadly across tokens irrelevant to the current inference step, preventing the model from focusing sufficiently on the tokens that determine correctness. The second is **Attention Drift**: As generation progresses, the model's attention is increasingly captured by the most recently generated code segments, causing the initial plan to be marginalized and critical constraints to be overlooked. These issues severely impair the efficacy of models operating under the plan-then-code paradigm.

To address these issues, we propose **PADA** (**P**erturbation-**V**erified **A**ttention **D**istillation and Dynamic **A**lignment). We first record the attention matrices of the student and teacher models, then extract high-attention key tokens from both, using perturbation analysis to assess their importance. Consequently, we introduce *Distillation Information Density* (DID) to extract a semantically dense set of key tokens, serving as the basis for addressing attention dispersion. Then we introduce the *Dynamic Attention Alignment* employed during the training phase. This approach integrates Difficulty-Aware gating and a Sliding Window mechanism, thereby dynamically mitigating the issue of attention drift. The resulting model is denoted as **PADA-Coder**. The main contributions of this paper are summarized as follows:

- We investigate the Plan-then-Code paradigm from the perspective of attention mechanisms and identify a critical bottleneck termed Attention Allocation Imbalance, which consists of two aspects: *Attention Dispersion* and *Attention Drift*.

- We propose PADA, a unified framework to simultaneously mitigate attention dispersion and drift. By constructing Maximum-DID Matrix via Perturbation-Verified Attention Distillation and a Dynamic Attention Alignment strategy, PADA dynamically calibrates the model's focus, ensuring strict adherence to the plan throughout the generation process.

- We evaluate PADA with two teacher models and three student models across seven benchmarks covering different task difficulty levels. PADA improves Pass@1 by up to 16.7% across all settings, outperforming all SOTA methods. Notably, with PADA, small-scale LLMs can achieve performance comparable to, or even surpassing ultra-large scale LLMs (Gemini-2.5-pro).

## 2. Related Work

### 2.1. Attention Distillation

MiniLM (Wang et al., 2020) first proposed that model capability can be improved through attention distillation. A2D

(Jin et al., 2024) introduces an alignment mechanism to determine the optimal correspondence between student and teacher attention heads. LeaF (Guo et al., 2025) uses Causal Attention Distillation to enhance the model's code generation ability. However, merely replicating the teacher's attention patterns, either fully or partially, does not suffice to resolve the issue of *attention drift*.

### 2.2. Attention guidance

SPA (Tian & Zhang, 2025) guides LLMs to maintain attention on user intent by anchoring key prompt tokens. SyntaGuid (Gesi & Ahmed, 2024) introduces SyntaGuid that mitigates attention bias in Transformer models by steering attention weights. Self-Anchor (Zhang et al., 2025) decomposes reasoning trajectories into formally structured subplans and constrains the model's attention to the most pertinent intermediate inference steps. However, these methods are either based on rigid rules or lack a mechanism for aligning key tokens, thus failing to truly address the issue of *attention dispersion*.

### 2.3. Chain-of-Thought code generation

Several methods have been proposed to enhance the performance of LLMs in code generation tasks through Chain-of-Thought. Zero-shot reasoning (Kojima et al., 2022) leverages prompts such as "let's think step by step" to guide the model through the process of code generation. Self-refine (Madaan et al., 2023) empowers models to iteratively refine their code by providing self-feedback. Least-to-most (Zhou et al., 2023) prompting enables complex reasoning by breaking down tasks into sequential sub-tasks. Chain-of-verification (Dhuliawala et al., 2024) reduces hallucination in generated content by requiring the model to verify and refine its outputs through a chain of validation steps. LDB (Zhong et al., 2024) leverages step-by-step runtime execution verification. LPW (Lei et al., 2025) integrates a pre-generation planning phase where the model outlines the solution before producing the code.

## 3. Methodology

In "Plan-then-Code" framework, models often generate incorrect code due to **Attention Allocation Imbalance**, which results in misalignment with the plan. Specifically, this imbalance manifests in two forms: (1) **Attention Dispersion** and (2) **Attention Drift**. To effectively address these issues, we propose a novel framework: (1) We extract key tokens of the plan from both the teacher and student models. Then we construct a maximum information density matrix with **Perturbation-Verified Attention Distillation** to mitigate *attention dispersion*. (2) We propose a **Dynamic Attention Alignment** training strategy, combining a progress-aware sliding window mechanism and a difficulty-aware gating

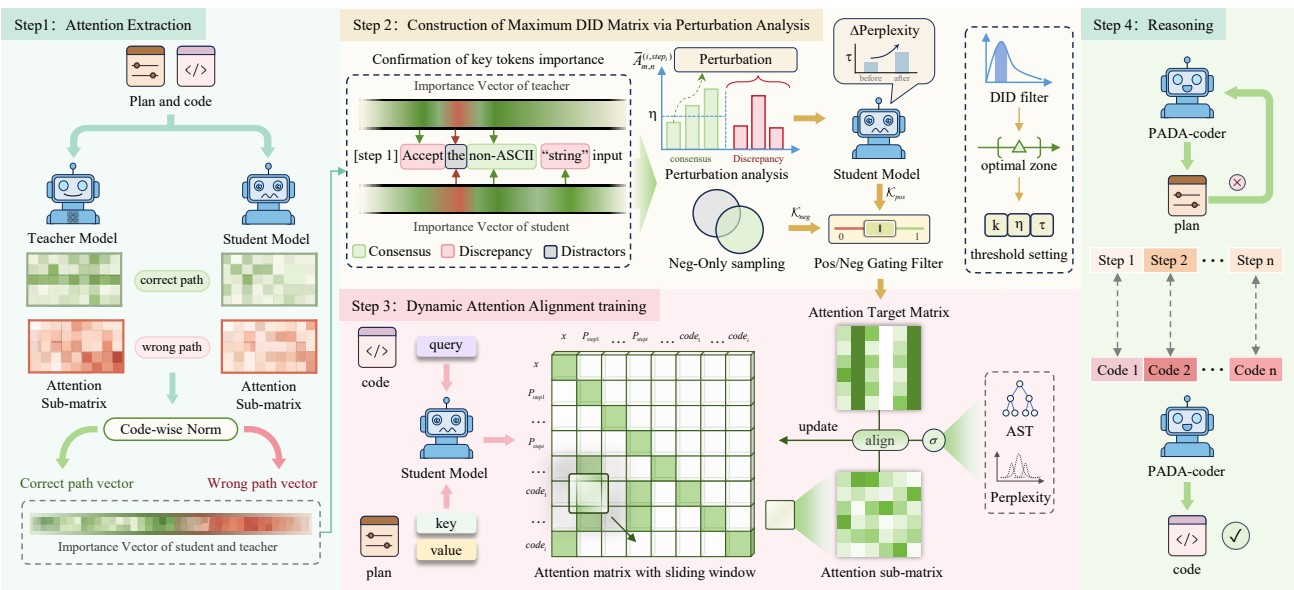

*Figure 2.* **Method Overview**. Our framework comprises three steps:(1) **Attention Extraction:** we extract the attention matrices from both teacher and student models corresponding to correct and erroneous outputs, followed by Code-wise Aggregation to derive importance vectors. (2) **Construction of Maximum DID Matrix via Perturbation Analysis:** Based on DID, we get $k$, $\eta$ and $\tau$. the top-$k$ key tokens are selected and categorized into consensus, divergence, and Distractors based on the ranked attention scores. The consensus and divergence undergo perturbation analysis to yield the key tokens set $K_{pos}$, while the Distractors are processed via neg-only sampling to obtain the $K_{neg}$. Then we construct the Maximum-DID Attention Target Matrix with these sets. (3) **Dynamic Attention Alignment training:** The student model updates its attention matrix via a sliding window and gating mechanism to align with the target matrix, thereby enhancing generation accuracy.

system to dynamically adjust alignment intensity and resolve *attention drift*. A brief illustration of our method is provided in Figure 2.

### 3.1. Training Data Construction and Attention Extraction

#### 3.1.1. CONSTRUCTION OF TRAINING DATA

We input problem x into teacher model and obtain the plan $p = \{p_{step_1}, \ldots, p_{step_n}\}$ and verification report $v$ through the Dual-Stage Plan Construction Strategy (see appendix A). The teacher model partitions the plan into a sequence of steps, where $step_i$ corresponds to the token subsequence for the $i$-th plan step.

Fixing $p$ and $v$, we conduct a generation-and-verification cycle to produce code. We retain the correct code as positive sample ($y^+$), Incorrect code from the first failed attempt as Negative Samples ($y^-$). Finally, we align the code $y = [y^{(1)}, \ldots, y^{(n)}]$ with plan $p$. Samples that cannot be reliably aligned are filtered out. Consequently, each training instance is formulated as $(x, p, v, y)$, where $y = y^+ \cup y^-$.

#### 3.1.2. ATTENTION SET CONSTRUCTION

Recent research shows that models assign higher attention scores to key information during inference, and that stronger

focus on this information generally improves generation quality (Li et al., 2025b). We therefore define the tokens receiving higher attention scores in positive and negative samples, respectively, as Key Tokens. We first precisely characterize the *Intrinsic Attention Distributions* of LLM. For the $h$-th attention head in the $\ell$-th layer, the attention score $\mathbf{A}^{(\ell,h)}$ are given by:

$$\mathbf{A}^{(\ell,h)} = \text{softmax}(\frac{\mathbf{QK}^\top}{\sqrt{d}}) \in \mathbb{R}^{T \times T}, \quad (1)$$

The sub-matrix for "Code $i \to$ Plan step $i$" is defined as:

$$\mathbf{A}^{(i,step_i,\ell,h)} \triangleq \mathbf{A}^{(\ell,h)}[i, step_i] \in \mathbb{R}^{|y^{(i)}| \times |p_{step_i}|}. \quad (2)$$

To quantify the contribution of each token in the plan step $p_{step_i}$ to the generation of code segment $y^{(i)}$, we compute a global Importance Vector $\mathbf{v}^{(i,step_i)}$ by averaging the raw attention scores across all layers ($L$), heads ($H$), and generated code tokens ($|y^{(i)}|$):

$$\mathbf{v}_n^{(i,step_i)} = \frac{1}{L \cdot H \cdot |y^{(i)}|} \sum_{l=1}^{L} \sum_{h=1}^{H} \sum_{m=1}^{|y^{(i)}|} \mathbf{A}_{m,n}^{(i,step_i,l,h)}. \quad (3)$$

where $\mathbf{v}_n^{(i,step_i)}$ represents the aggregated attention scores of the $n$-th plan token. Finally, we select the Top-$k$ tokens

with the highest scores from $\mathbf{v}^{(i,step_i)}$ as the initial focus set $\mathcal{K}_{raw}$.

LLMs allocate many attention scores to semantically null tokens like spaces or punctuation(Razzhigaev et al., 2025). To address this, we use a Conditional Expansion strategy. For each token $t_i \in \mathcal{K}_{raw}$, its expansion window $W(t_i)$ includes neighboring tokens if $t_i$ is a separator. The final key token set is the union of all windows: $\mathcal{K} = \bigcup_{t_i \in \mathcal{K}_{raw}} W(t_i)$.

We apply the method above separately to both the teacher and student models to obtain the key tokens focused on by each. We then obtain the Teacher key tokens Set ($\mathcal{K}_T$) and the Student key tokens Set ($\mathcal{K}_S$), where:

$$\mathcal{K}_T = \mathcal{K}_T^+ \cup \mathcal{K}_T^-, \quad \mathcal{K}_S = \mathcal{K}_S^+ \cup \mathcal{K}_S^-. \quad (4)$$

### 3.2. Perturbation-Verified Attention Distillation and Maximum-DID Matrix Construction

To mitigate *attention dispersion*, we explored attention distillation. However, traditional attention distillation forces students to fully or partially match the teacher's attention, ignoring the student's own priors and potentially transferring unnecessary or noisy alignment signals (Li et al., 2024). We refer to this issue as **Teacher Imitation Bias**. To address this limitation and identify the most important key tokens, we introduce Perturbation-Verified Attention Distillation.

#### 3.2.1. DISTILLATION INFORMATION DENSITY (DID)

Constructing an optimal target matrix involves a fundamental trade-off. Since the Transformer's attention mechanism employs Softmax normalization, the total attention budget is fixed. Consequently, selecting a larger number of distillation key tokens inevitably reduces the attention score allocated to each key token, potentially leading to insufficient focus on critical logical nodes. Conversely, selecting too few key tokens risks losing necessary contextual constraints.

To achieve an optimal trade-off, we introduce **Distillation Information Density (DID)**. This metric aims to identify a key token set that maximizes the information density of Attention Target Matrix. By concentrating the limited attention budget on this high-density set, we maximize code generation accuracy while avoiding attention dilution.

First, we employ the following formula to characterize the *Average Attention Efficiency*, measuring whether the model can distinguish each key token from the context noise:

$$\bar{a}(\rho, t) = \frac{1}{1 + \left(\frac{1}{\text{erf}(\kappa \rho L)} - 1\right) \cdot \exp\left(-\frac{\eta t}{(\rho L)^\beta}\right)}, \quad (5)$$

then, we model the Distillation Information Density using the following fitting function:

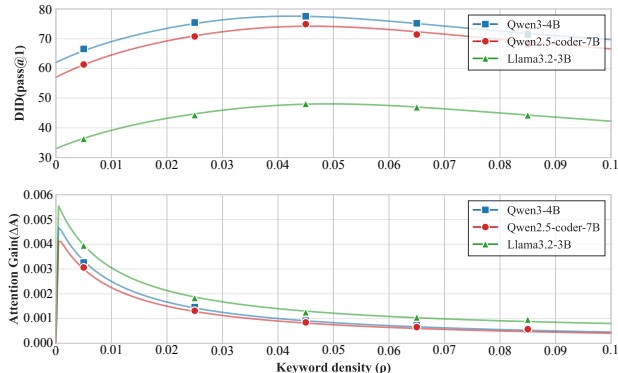

*Figure 3.* We conducted experiments with token retention ratios of $[0.5\%, 2.5\%, 4.5\%, 6.5\%, 8.5\%]$ across three student models. The data points represent our experimental data, and the curve represents our function fitting result. The relationship between DID and key token density, showing an Inverted U-shape.

$$DID(\rho) = \alpha \cdot \frac{\ln(\rho L)}{\rho L} \cdot \bar{a}(\rho, t), \quad (6)$$

$$pass@1 \approx \mathcal{F}\big(\text{DID}(\rho)\big) \cdot \mathbb{1}\big(\mathcal{I}(\rho) \geq \mathcal{I}_{critical}\big), \quad (7)$$

where $\rho$ denotes the proportion of key tokens within a full context of length $L$, $t$ represents the training steps, and $\alpha$, $\eta$, and $\beta$ are coefficients. Detailed rules for coefficients selection are provided in Appendix C.4. As illustrated in Figure 3, the optimal region is observed at a density of approximately **4%–6%**. Based on this zone, we dynamically adjust the subsequent selection of the Top-$k$ value for key tokens as well as the sampling perturbation threshold $\eta$ and the perplexity increment threshold $\tau$ to ensure the number of key tokens remains within this optimal region. The detailed proof is provided in Appendix C.

#### 3.2.2. DETERMINE KEY TOKEN IMPORTANCE VIA PERTURBATION ANALYSIS

We define the following subsets:

**Consensus Set:** $\mathcal{C}^{(i)} = \mathcal{K}_T^{(i)} \cap \mathcal{K}_S^{(i)}$. This set contains tokens deemed important by both models.

**Disagreement Set:** $\mathcal{D}^{(i)} = (\mathcal{K}_T^{(i)} \cup \mathcal{K}_S^{(i)}) \setminus \mathcal{C}^{(i)}$. This set contains tokens where the models diverge, representing uncertain information value.

To distill high-value key tokens from these sets, we employ **Perturbation analysis**. We propose *Perplexity Increment* ($\Delta$PPL) as the filtering metric. This metric quantifies the information density of each key token by measuring the extent to which masking a specific token increases the model's perplexity. Perplexity (PPL) is a standard metric for evaluating

an LLM's predictive capability given a context, essentially characterizing the model's uncertainty regarding the true data distribution(Mora-Cross & Ramírez, 2024). Given the context $p$, the PPL for $y_i$ is:

$$\text{PPL}(y_i|p) = \exp\left(-\frac{1}{L}\sum_{i=1}^{L}\log p(y_i|p)\right). \qquad (8)$$

For any candidate token $t$, we use mask perturbation to measure its impact on the generation of the code. Let $p_{\backslash t}$ denote the plan text where token $t$ is masked. Freezing the model parameters, we calculate the $\Delta PPL$ for the $i$-th correct code segment $y^{+(i)}$ under the perturbed context to measure the importance of this content:

$$\Delta\text{PPL}_{i,t} = \text{PPL}(y^{+(i)} \mid p_{\backslash t}) - \text{PPL}(y^{+(i)} \mid p). \quad (9)$$

A larger $\Delta PPL_{i,t}$ indicates that masking the token significantly increases the difficulty of predicting the code segment, implying that the token is critical for generation.

**For the Consensus Set $\mathcal{C}^{(i)}$:** Based on the student importance vector $\mathbf{v}^{(i,step_i)}$, we perform **sampling-based perturbation** over tokens in $\mathcal{C}^{(i)}$ whose scores satisfy $\mathbf{v}_j^{(i,step_i)} < \eta$, yielding a sampled perturbed subset $\mathcal{R}^{(i)} \subseteq \mathcal{C}^{(i)}$.

**For the Disagreement Set $\mathcal{D}^{(i)}$:** We perform **Full Perturbation** by computing $\Delta PPL$ for every token $t \in \mathcal{D}^{(i)}$.

We apply a threshold filter $\Delta PPL_{i,t} > \tau$ for each token $t \in \mathcal{D}^{(i)} \cup \mathcal{R}^{(i)}$ to identify a subset of most important key tokens. This procedure yields the final set of key tokens of positive samples, denoted by $\mathcal{K}_{\text{pos}}$. Then we define the set of key tokens of the negative samples as $\mathcal{K}_{\text{neg}} = \mathcal{K}_S^- \cup \mathcal{K}_T^- \setminus \mathcal{K}_{\text{pos}}$.

### 3.2.3. ATTENTION TARGET MATRIX CONSTRUCTION

Finally, we apply a positive/negative (Pos/Neg) gating filter to construct Attention Target Matrix $M^{target}$. In practice, utilizing binary supervisory signals to guide attention patterns serves as an effective inductive bias, despite their theoretical misalignment with the softmax function. This approach provide a sharpened, high-contrast objective that effectively guides the model to attend to key tokens, that is well-established and proven effective in previous works (Gesi & Ahmed, 2024; Deshpande & Narasimhan, 2020). For any plan token $T_n$ at column index $n$.

$$M_{\cdot,n}^{target} = \mathbb{I}(T_n \in \mathcal{K}_{\text{pos}}) \cdot 1 + \mathbb{I}(T_n \in \mathcal{K}_{\text{neg}}) \cdot 0. \quad (10)$$

### 3.3. Dynamic Attention Alignment Training

To mitigate the issue of **Attention drift**, we design a Dynamic Attention Alignment training strategy incorporating a sliding window and difficulty gating mechanisms.

### 3.3.1. PROGRESS-AWARE SLIDING WINDOW

During the training phase, requiring the model to attend to the plan $p$ while learning each code segment $y^{(k)}$ is suboptimal. To address this, we propose a **Progress-Aware Sliding Window**. For each code segment $y^{(i)}$ implements the corresponding plan step $p_{step_i}$. When training on $y^{(i)}$, we construct a dynamic focus window $W_i$ that includes only the plan steps causally related to the current implementation:

$$W_k = \{p_{\text{step}_{i\text{-}2}},\ p_{\text{step}_{i\text{-}1}},\ p_{\text{step}_i}\}. \qquad (11)$$

In this manner, the sliding window moves across the plan in synchronization with the training progress $i$ of each sample (Li et al., 2026c).

### 3.3.2. DIFFICULTY-AWARE GATING MECHANISM

To prevent *Plan Over-Interpretation* on simple code segments, we introduce a Dual-gating system $\lambda_{\text{gate}}$ driven by structural complexity $S_{\text{AST}}$ (normalized AST depth and cyclomatic complexity) and model perplexity $S_{\text{PPL}}$ (on $y^+$). The gate dynamically regulates the attention loss weight:

$$\lambda_{\text{gate}} = \text{sigmoid}\left(\frac{\lambda \cdot S_{\text{AST}} + \ln(S_{\text{PPL}}) - \mu}{\mathcal{T}}\right), \qquad (12)$$

where $\lambda$ balances the scale between structural and statistical difficulty, $\mu$ serves as the difficulty threshold (centering the sigmoid), and $\mathcal{T}$ is a temperature parameter controlling the transition sharpness of the gate.

### 3.3.3. OPTIMIZATION OBJECTIVE

The final loss function comprises the language modeling loss $\mathcal{L}_{LM}$ and the attention alignment loss $\mathcal{L}_{attn}$. $\mathcal{L}_{attn}$ is computed as the normalized sum of squared Frobenius differences between the student and target attention matrices across selected layers $l$ and heads $h$ (see Appendix J for specific target layers and target heads). We define $\mathcal{L}_{attn}$ as:

$$\mathcal{L}_{attn} = \frac{1}{Z}\sum_{l=1}^{L}\sum_{h=1}^{H}\mathcal{I}(l,h)\cdot \\ \left\|\mathbf{A}^{(l,h)}[y^{(i)}, W_i] - \mathbf{M}^{target}\right\|_F^2, \qquad (13)$$

where $\mathcal{I}(l,h) \in \{0,1\}$ indicates the specific layers and heads targeted for alignment, and $Z = H_{sel} \times |y^{(i)}| \times |W_i|$ serves as the normalization factor. The total optimization objective is:

$$\mathcal{L}_{\text{total}} = \mathcal{L}_{\text{LM}} + \alpha(t) \cdot \lambda_{\text{gate}} \cdot \mathcal{L}_{\text{attn}}. \qquad (14)$$

$\alpha(t)$ is a decay coefficient that decreases linearly with training steps $t$.

*Table 1.* Comparative performance of PADA and other approaches across multiple benchmark datasets. APPS-I, APPS-V, and APPS-C represent the APPS dataset for Introductory, Interview, and Computational problem categories respectively. LCB refers to the LiveCodeBench v5 dataset. TM

| Model | Method | easy | | medium | | difficult | | | Average |
|---|---|---|---|---|---|---|---|---|---|
| | | HumanEval | MBPP | MBPP+ | APPS-I | APPS-V | APPS-C | LCB | |
| Qwen3-4B | Base | 87.8 | 79.6 | 70.5 | 61.3 | 50.7 | 32.7 | 53 | 62.2 |
| | SPA | 92.7 | 84.1 | 72.2 | 68 | 52 | 34 | 59.1 | 66.0 |
| | LPW | 95.1 | 91.1 | 83.7 | 77.3 | 62.4 | 49.3 | 69.5 | 75.5 |
| | LeaF | 93.3 | 85.7 | 74.9 | 69.3 | 57.6 | 38 | 61.6 | 68.6 |
| | PADA (w\o) TM | 95.7 | 89.9 | 82.2 | 77.6 | 63.5 | 47.2 | 71.3 | 75.3 |
| | PADA | 96.8 | 93.2 | 85.5 | 79.3 | 65.2 | 50.4 | 74.4 | 77.8 |
| Qwen2.5-coder-7B | Base | 89 | 75.9 | 66.4 | 59.3 | 47.3 | 24.7 | 35.4 | 56.9 |
| | SPA | 92.1 | 81 | 75.4 | 62.4 | 48.8 | 29.3 | 43.2 | 61.7 |
| | LPW | 97.6 | 89.7 | 86.7 | 72.7 | 52.7 | 40.7 | 53 | 70.4 |
| | LeaF | 94.5 | 84.8 | 79.4 | 66.7 | 50.2 | 39.3 | 47 | 66.0 |
| | PADA (w\o) TM | 95.1 | 88.8 | 83.6 | 72.6 | 58 | 44 | 55.4 | 71.1 |
| | PADA | 95.7 | 91.1 | 85.2 | 76 | 61.3 | 46.7 | 59.1 | 73.6 |
| Llama3.2-3B | Base | 63.4 | 47.5 | 30.8 | 37.3 | 25.3 | 10.7 | 18.3 | 33.3 |
| | SPA | 68.9 | 56 | 41.4 | 40 | 27.8 | 12.7 | 21.3 | 38.3 |
| | LPW | 67.6 | 52.9 | 33.6 | 40.7 | 28.2 | 8.7 | 16.5 | 35.5 |
| | LeaF | 71.3 | 60.2 | 44.6 | 43.3 | 28.7 | 14 | 23.8 | 40.8 |
| | PADA (w\o) TM | 72 | 64.6 | 48.6 | 45.3 | 30.7 | 17.3 | 26.2 | 43.5 |
| | PADA | 75.6 | 66.3 | 50.6 | 49.3 | 34.7 | 20.7 | 31.1 | 46.9 |

*Table 2.* Comparative performance of PADA-Coder and ultra-large scale LLMs across multiple benchmark datasets.

| Tasks | GPT-5.2 | deepseek-v3.2 | Gemini-2.5-pro | PADA-Llama3.2 | PADA-Qwen2.5 | PADA-Qwen3 |
|---|---|---|---|---|---|---|
| APPS-I | 71.3 | 68.7 | 72.7 | 49.3 | 76 | **79.3** |
| APPS-V | 53.1 | 48.6 | 57.3 | 34.7 | 61.3 | **65.2** |
| APPS-C | 38.7 | 35.3 | 40.7 | 20.7 | 46.7 | **50.4** |
| HumanEval | 96.3 | 97 | **97** | 75.6 | 95.1 | 96.8 |
| MBPP | 88.5 | 86.4 | 90.9 | 70.3 | 91.1 | **93.2** |
| MBPP+ | 82.2 | 80.5 | **87** | 50.6 | 85.2 | 85.5 |
| LiveCodeBench | 68.3 | 64.3 | 71.3 | 31.1 | 63.4 | **74.4** |
| Average | 71.2 | 68.7 | 73.8 | 47.5 | 74.1 | **77.8** |

# 4. Experiments

In this section, we conduct extensive experiments to answer the following four core research questions (RQs):

- **RQ1 (Main Results):** How does PADA-Coder perform on mainstream code generation benchmarks?

- **RQ2 (Plan Compliance):** Given a reference plan, can PADA-Coder translate natural language logic into code more accurately than closed-source models?

- **RQ3 (Ablation & Analysis):** What are the individual contributions of the proposed attention alignment, sliding window, and perturbation-based analysis mechanisms to the final performance?

- **RQ4 (Attention Allocation):** Does the proposed attention alignment method effectively direct the model's attention to more appropriate locations?

**Datasets:** We collected 4,000 samples from the APPS training set to serve as our training dataset. For benchmarking, we utilized the test splits of APPS (Introductory/Interview/-Computational) (Hendrycks et al., 2021), as well as HumanEval (Chen et al., 2021), MBPP (+) (Austin et al., 2021; Liu et al., 2023), and LiveCodeBench v5 (Jain et al., 2025). They were graded by difficulty level.

**Baselines:** We carefully selected baseline models from four distinct categories to conduct a comprehensive evaluation:

- **Attention Distillation Methods: LeaF** identifies attention discrepancies between teacher and student model. It employs pruning and mixed training strategies to help student correct erroneous attention positions.

- **Agent-based Methods: LPW** improves code through a "Plan-Code-Reflect" iterative paradigm.

- **Attention Guidance Methods: SPA** leverages explicit

attention guidance during the reasoning process to enhance the model's understanding of code problems.

- **ultra-large scale LLMs:** We include recently released ultra-large scale LLMs, including **Gemini-2.5-pro** (Group, 2023) (Gemini-3-pro API was unavailable at the time of submission), **GPT-5.2** (OpenAI, 2023), and **DeepSeek-v3.2** (DeepSeek-AI, 2024). We prompted these models to directly output solution code using the problem statement and public test cases as input.

**Implementation Details** For knowledge distillation, we served Qwen3-32B as the teacher model for Qwen3-4B (Yang et al., 2025) and Qwen2.5-7B-coder-instruct (Hui et al., 2024), while Llama3.3-70B-instruct was used for Llama3.2-3B-instruct (Team, 2024). We use **pass@1** for our evaluation. For **LeaF**, we used the same teacher-student model pairs and the same training dataset as in our method for a fair comparison. For **LPW**, we adopt the optimal iteration settings specified in the original paper, specifically performing 12 iterations. For **SPA**, we followed the optimal rule selection method described in the original paper. For **Closed-source LLMs**, we utilized their official APIs for generation.

## 4.1. Main Result (RQ1)

Our method achieved the most significant improvements across various models and datasets, especially on difficult tasks. On **APPS-Computational** and **LiveCodeBench**, Qwen2.5-coder-7B-instruct achieved an average improvement of **22.9%**. Notably, the untrained Llama3.2-3B-instruct exhibited significant instruction-following issues. Consequently, our method demonstrated a remarkable advantage on Llama3.2-3B-instruct, achieving the highest relative improvement rate of **40.8%**.

**Compared with SPA:** Although SPA employs rule-based guidance for anchoring, it lacks flexibility. While it improves problem understanding, it often leads to code errors by ignoring specific constraints.

**Compared with LPW:** Although scaling LPW to 12 iterations (LPW@12) achieves slightly higher accuracy on certain benchmarks (e.g., HumanEval and MBPP+)) for Qwen2.5-coder-7B, it incurs prohibitive token consumption (detailed in Appendix B.3). More importantly, relying on extensive execution feedback inevitably lengthens the context, which exacerbates the risk of Attention Allocation Imbalance. For small-scale LLMs like Llama3.2-3B-instruct, this elongated prompt context actually degrades performance, demonstrating that prompt engineering alone is insufficient to precisely guide code generation.

**Compared with LeaF:** While LeaF can identify important key tokens, it still suffers from attention drift. We observed that code generated by LeaF often contained errors in the later stages of generation, neglecting critical constraints.

To ensure a fair comparison, our method achieved the best results even without a teacher model. This proves the effectiveness of our perturbation method in mining key tokens that are truly critical for code generation.

Furthermore, PADA achieves competitive or even superior performance across multiple benchmarks over ultra-large scale LLMs. This suggests that optimizing the reasoning process via precise attention guidance can effectively bridge the massive gap in parameter scale.

## 4.2. Evaluation with Reference Plans(RQ2)

To disentangle the effects of plan quality from code generation capability, we utilized Claude-sonnet-4.5 (Anthropic, 2024) to produce accurate reference plans for the test set. Because Claude-Sonnet-4.5 achieves near-oracle correctness in our verification pipeline—whereas small-scale LLMs often struggle with the most difficult problems—this approach ensures that our evaluation of plan-following capability is not bottlenecked by inherent planning limitations. By feeding these plans as prompts into different models, we assessed their proficiency in converting reference plans into correct code. As shown in Table 3, even with a correct plan, ultra-large scale LLMs may still produce incorrect code.

*Table 3.* Comparative evaluation of PADA-Coder using Qwen3 as the backbone model and ultra–large scale LLMs across multiple benchmark datasets with reference plans.

| Model | APPS | HumanEval | MBPP+ | Avg |
|---|---|---|---|---|
| deepseek-v3.2 | 66.8 | 95.7 | 87.4 | 83.3 |
| Gemini-2.5-pro | 71.3 | 99.3 | 91.0 | 87.2 |
| gpt-5.2 | 68.7 | 99.3 | 89.2 | 85.7 |
| Base | 60.1 | 94.5 | 86.7 | 80.4 |
| ours | 75.7 | 98.2 | 91.7 | 88.5 |

**Performance on Simple Tasks:** PADA-Coder slightly lagged behind Gemini-2.5-Pro and GPT-5.2 on HumanEval, likely because Attention Allocation Imbalance is less pronounced in ultra-large LLMs handling easy, low-complexity, short-context tasks.

**Advantage on Difficult Tasks:** In comparison, PADA-Coder performs better on both MBPP+ and APPS datasets, especially on the APPS. Since APPS problems are more difficult and lengthy, with correspondingly longer plans and tests, LLMs are more prone to encounter **Attention Allocation Imbalance**. In contrast, PADA-Coder, trained to consistently focus on plan constraints, excels in these

challenging, long-context scenarios.

## 4.3. Ablation Studies (RQ3)

To verify the effectiveness of each component of PADA-Coder, we conducted ablation studies on the APPS and HumanEval datasets using Qwen3-4B. The detailed experimental results are presented in Table

*Table 4.* Experiment on Ablation Study.

| Variants | APPS | HumanEval |
|---|---|---|
| Base | 49.1 | 87.8 |
| PADA | 65.1 | 96.8 |
| w/o Sliding Window | 61.8 | 93.2 |
| w/o perturbation | 63.1 | 95.1 |
| w/o attention alignment | 57.8 | 93.9 |
| w/o difficulty gating | 65.1 | 96.3 |

**Removal of Attention Alignment Module:** This is equivalent to performing only standard fine-tuning (SFT). This variant resulted in the most significant performance drop. Because our training targets attention alignment, the model does not need the large data volumes usually required for SFT. However, SFT on only small-scale data makes it hard for the model to learn complex code logic.

**Removal of Sliding Window Mechanism:** Without the sliding window, the model is required to attend to the entire plan simultaneously. This introduces excessive irrelevant information during the inference of code segments.

**Removal of Perturbation Strategy:** This equates to forcibly aligning the student model's attention pattern to match the teacher's. This leads "Teacher Imitation Bias", resulting in suboptimal alignment.

**Removal of difficulty gating:** As shown in Table 4, removing difficulty gating yields almost no change in accuracy. Rather than directly boosting correctness, its primary role is to prevent "Plan Over-Interpretation" on simple code segments. By avoiding over-allocation of attention to straightforward logic, it significantly reduces token consumption and inference overhead, achieving a superior cost-performance balance (see Appendix E) .

## 4.4. Visualization of Attention Patterns(RQ4)

To validate the improvement, we visualize the average attention scores allocated to key tokens and the total plan across all layers and heads during inference. We established the test set by selecting 50 samples with more plan steps from HumanEval, MBPP+, and APPS respectively, and extracting the key tokens via perturbation analysis. We divided this experiment into two parts.

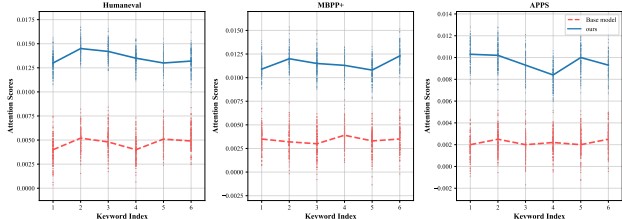

*(a)* The attention scores of the first 6 key tokens

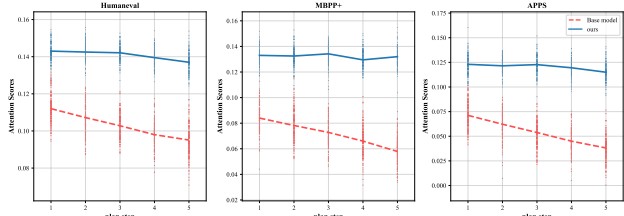

*(b)* the overall attention allocation trend towards the plan content during the first 5 planning steps of generating code

*Figure 4.* Visualization of attention patterns comparing PADA-Coder (Blue Line) and Base Model (Red Dashed Line).

As shown in Figure 4a, our method consistently maintained higher attention scores than the base model across all datasets, particularly on the more challenging APPS dataset. The base model consistently assigns low attention scores to plan key tokens, indicating a potential failure to allocate sufficient attention. In contrast, our method successfully maintained attention at a high level. This result shows that our method mitigating the issue of attention distraction.

As shown in Figure 4b, the baseline model showed a clear downward trend in attention scores as the coding progressed, indicating a significant **attention drift** problem. Conversely, our method maintained a stable and high-intensity attention allocation on plan across all steps.

In conclusion, the visualized attention allocation strongly proves that our method mitigates the problem of *Attention Allocation Imbalance* during code generation, ensuring consistency between the plan and the generated code.

## 4.5. Evaluation on Mathematical Reasoning Tasks

*Table 5.* Evaluation of PADA built on Qwen3-4B and Llama3.2-3B across mathematical reasoning benchmarks. PS represents Plan-then-Solve (Wang et al., 2023).

| Model | method | GSM8K | MATH-500 |
|---|---|---|---|
| Qwen3-4B | PS | 86.7 | 82.4 |
| | SFT | 86.9 | 83 |
| | PADA | 88.1 | 85.4 |
| Llama3.2-3B | PS | 62.2 | 42.5 |
| | SFT | 62.9 | 43.4 |
| | PADA | 71.3 | 46.4 |

Although our focus in this work is on code generation, we are equally interested in whether PADA can be applied to other reasoning tasks. Thus, we evaluate PADA on mathematical reasoning benchmarks, including GSM8K (Cobbe et al., 2021) and MATH (Lightman et al., 2024).

As shown in Table 5, PADA enhances the base model's performance in both benchmarks. Although the observed performance is not as significant as in the domain of code generation, we conjecture that for mathematical reasoning—where internal reasoning ability is crucial—simply increasing attention to key tokens offers limited efficacy. We additionally observed that PADA yielded the largest performance improvement when applied to Llama3.2-3B on GSM8K. We conjecture that this may be because GSM8K mostly consists of text questions, which facilitates the identification and mapping of key tokens. In contrast, Llama3.2-3B-instruct appears more susceptible to attention imbalance during the selection of these key tokens. Nevertheless, an appealing direction for future work is to explore how PADA can be further enhanced to support a wider spectrum of reasoning tasks beyond code generation.

### 4.6. Evaluation on other teacher models

Research has shown that in the field of coding, it is possible to attempt code LLM self-alignment without human annotation(Wei et al., 2024). By drawing on this idea, we observed that in our framework, the teacher model primarily serves as a source of supplementary key token information rather than being an indispensable component for performance improvement. This led us to explore whether the teacher model can be replaced with models of similar parameter sizes or even with large closed-source models.

In this experiment, we compared the performance of Qwen2.5-7B-coder with similar-scale LLM, larger-scale LLM, and a closed-source LLM as the source of supplementary information. We evaluated the performance of these setups across three datasets: HumanEval, MBPP+, and APPS. We use prompt engineering for closed-source LLM to select what it deems to be important key tokens.

*Table 6.* Comparison of student model performance under various teacher models across multiple code generation benchmarks.

| teacher model | HumanEval | MBPP+ | APPS |
|---|---|---|---|
| Base | 95.1 | 83.6 | 58.1 |
| Qwen3-4B | 95.1 | 84.2 | 58.4 |
| Qwen3-32B | 95.7 | 85.2 | 61.3 |
| GPT-5.2 | 95.1 | 83.6 | 58.8 |

As shown in Table 6, larger-scale LLM (Qwen3-32B) yields the most substantial performance gains. Conversely, the similar-scale LLM (Qwen3-4B) offers negligible improve-

ment over the baseline. The explanation for why teachers with similar scores perform poorly is provided in the appendix F. While the closed-source model (GPT-5.2) aids in APPS, it underperforms on simpler tasks. We hypothesize this is because prompt-based methods only capture explicit semantic key tokens. However, we found that focusing on some implicit control signals (such as specific syntactic symbols) can improve model performance. However, it is difficult for open-source large models to provide guidance on such specific syntactic symbols.

Therefore, it is an interesting research direction to explore whether open-source large models can be used to guide attention distillation through certain means in the future.

## 5. Conclusion

In this paper, we introduce PADA, a novel framework to address the pervasive issue of Attention Allocation Imbalance in 'Plan-then-Code' paradigm of LLMs. Through perturbation analysis to construct Maximum-DID Matrix, we extract the key tokens most critical to the student model, thereby facilitating Dynamic Attention Alignment training to enhance the model's adherence to the plan. By leveraging PADA, the student model significantly enhances and sustains its focus on critical keywords during the inference phase. Empirical results demonstrate that our model achieves substantial improvements in Pass@1 accuracy across diverse benchmarks.

## Acknowledgments

This work is supported by National Natural Science Foundation of China No.62406057, the Fundamental Research Funds for the Central Universities No.ZYGX2025XJ042, the Noncommunicable Chronic Diseases-National Science and Technology Major Project No.2023ZD0501806, and the Sichuan Science and Technology Program under Grant No.2024ZDZX0011.

## Impact Statement

This paper presents a method for improving plan-following code generation in large language models. The proposed approach enables small-scale LLMs to achieve competitive or superior performance on complex coding tasks, potentially reducing computational costs and making advanced code generation more accessible in resource-constrained environments. However, more reliable code generation may also lower the barrier for deploying LLMs in safety-critical or malicious settings where generated code could contain undetected errors or be misused. We encourage users to apply the method with domain-specific validation, thorough testing of generated outputs, and appropriate human oversight.

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

# A. Preliminary Experiment

## A.1. Analyzing the Role of Plan in inference

We conducted a preliminary study to investigate the internal attention mechanism discrepancies between correct and incorrect code generations. Specifically, we analyzed the attention scores focused on the provided Plan tokens. We selected several questions from the APPS training set with similar lengths for testing. The attention scores are aggregated into sequential token blocks (bins of 50 tokens).

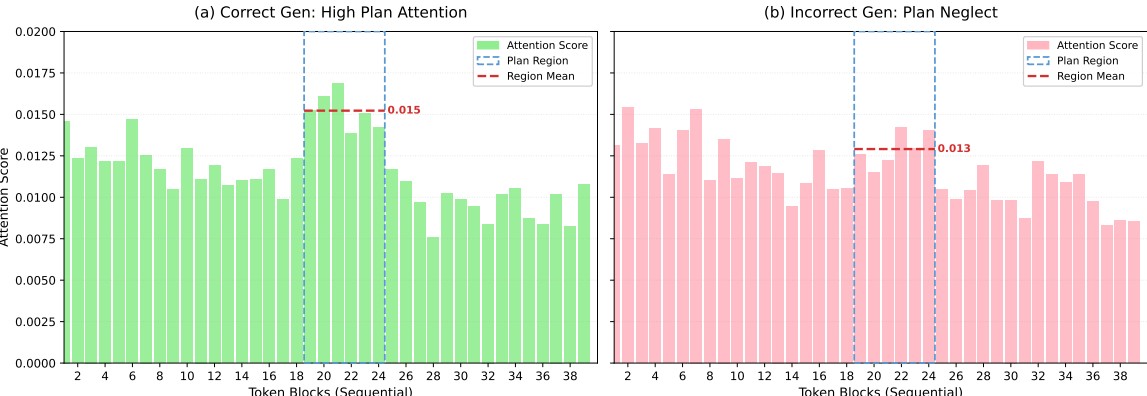

*Figure 5.* Comparative visualization of attention distributions during the generation process. The Plan Region is enclosed by a blue dashed box, with the red dashed line indicating the regional mean score. Specifically, the first bin would be abnormally high attributed to the attention sink phenomenon. Therefore, we removed the first bin to prevent it from affecting our observations.

In both correct (a) and incorrect (b) generations in Figure 5, the model inherently assigns higher attention scores to the plan compared to the other context, suggesting the model recognizes the plan's significance. However, the mean attention scores in the correct generation is significantly higher than in the incorrect one. This discrepancy suggests that while the model knows where to focus, it often fails to focus enough during erroneous generation, motivating our proposed method to explicitly guide and boost attention towards plan key tokens.

## A.2. Analyzing the attention allocation of LLMs in inference

Integrating dynamics-aware optimization strategies has proven highly effective (Wang et al., 2026). We investigated whether the model's attention allocation exhibits a systematic shift corresponding to the plan during the code generation process. Utilizing Qwen2.5-Coder-7B as the backbone, we fed the model with plans and their corresponding ground-truth code. By applying the attention aggregation method, we extracted the top 10% of tokens with the highest attention scores to identify key tokens and computed their mean attention scores. A representative case is presented for illustration.

---

**Attention Allocation study for APPS 2436: code segment 1**

[GEN_GLOBAL_PLAN]

[Algorithm] String Filtering + Reverse Comparison

[STEP:1] Take the input string `s` and normalize it by converting to lowercase for case-insensitive matching.

[STEP:2] Filter the normalized string to keep only alphanumeric characters, building a cleaned character list/sequence.

[STEP:3] Create the reversed version of the cleaned sequence (e.g., via slicing) to represent the palindrome counterpart.

[STEP:4] Compare the cleaned sequence to its reversed sequence and return `True` if they match, otherwise return `False`.

[STEP:5] Ensure the logic naturally treats an empty cleaned sequence as a valid palindrome and returns `True` in that case.

---

*Figure 6.* Visualization of attention allocation over plan tokens during the generation of the first code segment. The tokens receiving the highest attention scores are highlighted in red; for clarity, these tokens are displayed as complete words.

---

**Attention Allocation study for APPS 2436: code segment 2**

[GEN_GLOBAL_PLAN]

[Algorithm] String Filtering + Reverse Comparison

[STEP:1]  Take the input string `s` and normalize it by converting to lowercase for case-insensitive matching.

[STEP:2]  Filter the normalized string to keep only alphanumeric characters, building a cleaned character list/sequence.

[STEP:3]  Create the reversed version of the cleaned sequence (e.g., via slicing) to represent the palindrome counterpart.

[STEP:4]  Compare the cleaned sequence to its reversed sequence and return `True` if they match, otherwise return `False`.

[STEP:5]  Ensure the logic naturally treats an empty cleaned sequence as a valid palindrome and returns `True` in that case.

*Figure 7.* Visualization of attention allocation over plan tokens during the generation of the second code segment.

---

**Attention Allocation study for APPS 2436: code segment 3**

[GEN_GLOBAL_PLAN]

[Algorithm] String Filtering + Reverse Comparison

[STEP:1]  Take the input string `s` and normalize it by converting to lowercase for case-insensitive matching.

[STEP:2]  Filter the normalized string to keep only alphanumeric characters, building a cleaned character list/sequence.

[STEP:3]  Create the reversed version of the cleaned sequence (e.g., via slicing) to represent the palindrome counterpart.

[STEP:4]  Compare the cleaned sequence to its reversed sequence and return `True` if they match, otherwise return `False`.

[STEP:5]  Ensure the logic naturally treats an empty cleaned sequence as a valid palindrome and returns `True` in that case.

*Figure 8.* Visualization of attention allocation over plan tokens during the generation of the third code segment.

---

**Attention Allocation study for APPS 2436: code segment 4**

[GEN_GLOBAL_PLAN]

[Algorithm] String Filtering + Reverse Comparison

[STEP:1]  Take the input string `s` and normalize it by converting to lowercase for case-insensitive matching.

[STEP:2]  Filter the normalized string to keep only alphanumeric characters, building a cleaned character list/sequence.

[STEP:3]  Create the reversed version of the cleaned sequence (e.g., via slicing) to represent the palindrome counterpart.

[STEP:4]  Compare the cleaned sequence to its reversed sequence and return `True` if they match, otherwise return `False`.

[STEP:5]  Ensure the logic naturally treats an empty cleaned sequence as a valid palindrome and returns `True` in that case.

*Figure 9.* Visualization of attention allocation over plan tokens during the generation of the forth code segment.

> Attention Allocation study for APPS 2436: code segment 5
>
> [GEN_GLOBAL_PLAN]
>
> [Algorithm] String Filtering + Reverse Comparison
>
> [STEP:1]  Take the input string `s` and normalize it by converting to lowercase for case-insensitive matching.
>
> [STEP:2]  Filter the normalized string to keep only alphanumeric characters, building a cleaned character list/sequence.
>
> [STEP:3]  Create the reversed version of the cleaned sequence (e.g., via slicing) to represent the palindrome counterpart.
>
> [STEP:4]  Compare the cleaned sequence to its reversed sequence and return `True` if they match, otherwise return `False`.
>
> [STEP:5]  Ensure the logic naturally treats an empty cleaned sequence as a valid palindrome and returns `True` in that case.

*Figure 10.* Visualization of attention allocation over plan tokens during the generation of the fifth code segment.

Through our observation of a subset of samples, we found that the model tends to allocate more attention to the current plan and the previous two steps when generating the current code segment. To analyze this phenomenon more rigorously, we selected 200 samples from the APPS training set, each with 5 plan steps, and analyzed which plan steps the model tends to focus on while generating the current code segment. We use the mean attention score $\bar{a}(\delta)$ to represent Aggregated Attention Distribution over Relative Plan Steps, calculated as:

$$\bar{a}(\delta) = \frac{1}{N_\delta} \sum_{t=1}^{N_\delta} \alpha_{t,\delta}, \tag{15}$$

where $\alpha_{t,\delta}$ is the attention scores at step $t$, and $N_\delta$ is the varying sequence length (e.g., $N_0 = 5, N_{\pm 1} = 4$). The statistical results are shown in Figure 11.

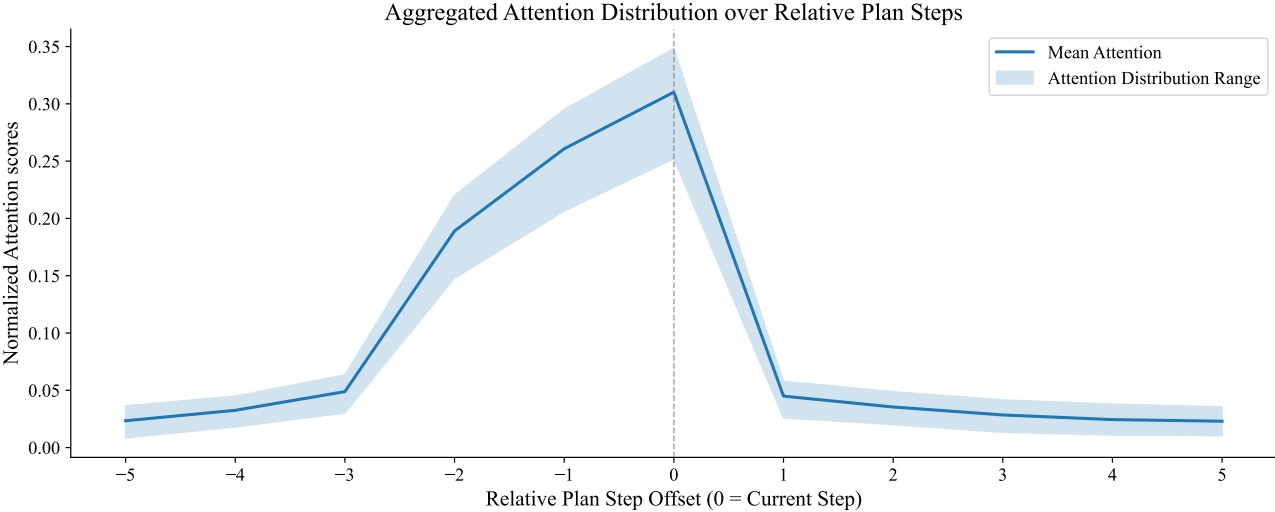

*Figure 11.* The visualization of mean Attention score relative to the current plan step. The solid line represents the mean attention score. We normalize the total attention by the decoding step frequency to eliminate length bias, thereby clearly visualizing the disparity in attention allocation across different plan steps. The shaded region indicates the min-max range (the full spread from minimum to maximum values) of the experimental distribution.

As illustrated in Figure 11, the attention distribution distinctly highlights a heavy concentration on the current step ($t = 0$) and its immediate predecessors ($t \in \{-1, -2\}$). This observation aligns with recent empirical findings suggesting that planning models primarily rely on current objectives and historical constraints to generate coherent code. Motivated by this,

we configure our sliding window $W_k$ as:

$$W_k = \{p_{\text{step}_{i-2}}, p_{\text{step}_{i-1}}, \ p_{\text{step}_i}\}. \tag{16}$$

In this manner, the sliding window can capture the most critical temporal context while filtering out irrelevant noise from distant steps.

## B. Analyzing the Role of Plan in PADA

### B.1. How does the correctness of a plan affect the model performance

We follow the "Plan-then-Code" paradigm, so we also need to conduct research on plans. Therefore, we designed the following experiment to compare the impact of correct and incorrect plans on model performance. The correct plans are produced by claude-sonnet-4.5, the incorrect plans are derived from validation failures. The table 7 shows the performance comparison of Qwen3-4B under correct and incorrect plans, specifically on the Humaneval, MBPP+, and APPS benchmark datasets.

*Table 7.* The performance of PADA-Coder given correct versus wrong plans

| Plan | Model | Humaneval | MBPP+ | APPS |
|---|---|---|---|---|
| Correct Plan | base | 94.5 | 85.7 | 64.9 |
| | PADA-Coder | 98.2 | 91.7 | 75.7 |
| Wrong Plan | base | 73.2 | 65.2 | 32.9 |
| | PADA-Coder | 71.4 | 64 | 30.8 |

As illustrated in table 7, we observe that PADA-Coder can better adhere to the correct plan, thereby generating accurate code. However, if the plan is incorrect, it may reinforce wrong reasoning, which proves that our method is closely related to the correctness of the generated plan. Notably, since we explicitly incorporated attention suppression on key tokens associated with erroneous reasoning paths, the model possesses a degree of robustness (Zhou et al., 2025b; Wang et al., 2025a) to selectively attenuate its focus on misleading key tokens within incorrect plans. The performance degradation caused by incorrect plan being more constrained than expected.

### B.2. Dual-Stage Plan Construction Strategy

In order to get high-quality correct plan, we proposed a Dual-Stage Plan construction Strategy, which is built on the LPW framework (Lei et al., 2025). The Plan Generation phase operates as a rigorous 'Plan-Verify-Refine' pipeline:

**Planning:** Utilizing the self-planning mechanism (Kojima et al., 2022; Zhu et al., 2025), LLM abstracts the input problem $x$ and decomposes it into a step-by-step, robust plan $p$.

**Execution Simulation:** To validate $p$, the model simulates its execution against the visible test set $T_v$. This process, denoted as $A(p, T_v)$, generates a comprehensive execution trace containing both intermediate states and final derived outputs ($to'_v$) for every test case.

**Output Consistency Check (First-pass Verification):** The system first verifies the terminal correctness. For each test case $t_v \in T_v$, it compares the derived output $to'_v$ with the ground truth $to_v$. The plan proceeds only if strict consistency is met ($\forall t_v, to'_v = to_v$).

**Intermediate Logic Validation (Second-pass Verification):** Crucially, successful output matching triggers a secondary review. The LLM scrutinizes the entire execution trace $A(p, T_v)$ to ensure the accuracy of all intermediate steps. This guarantees that the reasoning chain is sound, facilitating precise bug localization in later phases.

**Outcome:** Upon passing both verification stages, the validated plan and its verification report serve as the intended solution, guiding the subsequent code implementation phase.

As shown in Figure 12, we employ distinct strategies to acquire the initial plan $p$ depending on the phase mentioned above.

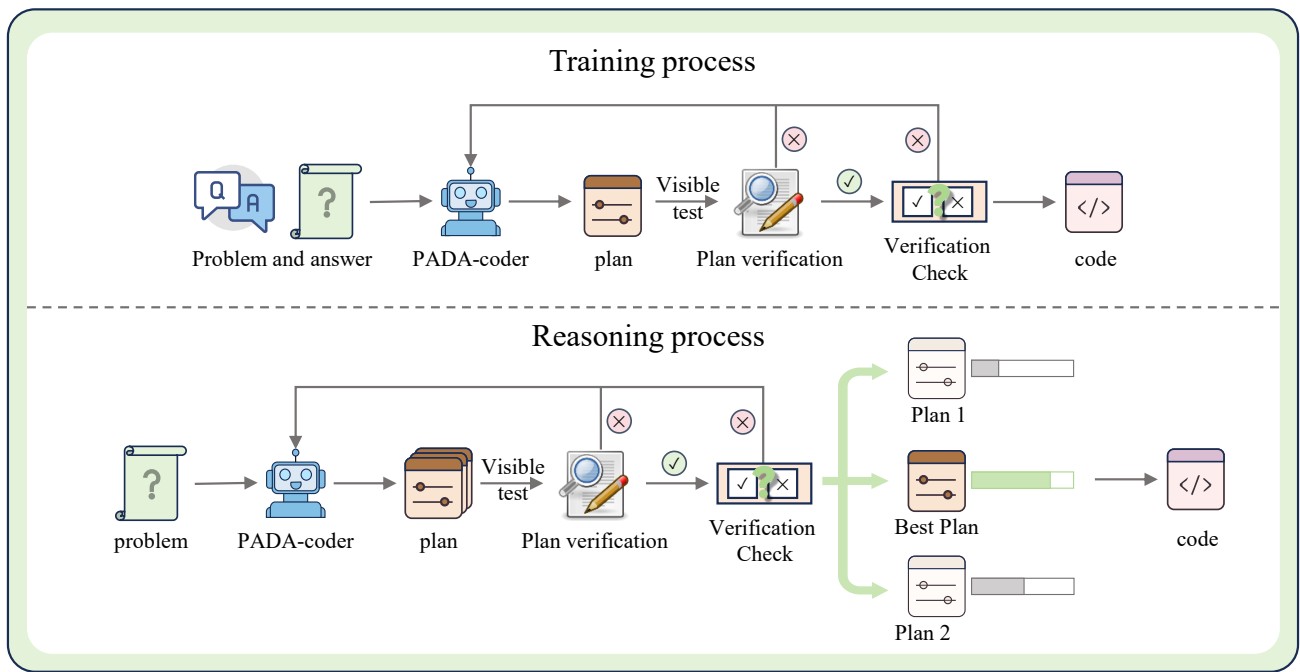

*Figure 12.* Overview of Dual-Stage Plan Construction Strategy, which constructs training data via answer-guided strategy and employs a multi-candidate selection strategy during inference to ensure high-quality planning.

**During Training (Data Construction):** We utilize the reference solutions (ground truth) to ensure optimal guidance. Specifically, before the planning step, we input the correct solution into the LLM to summarize the strategy used to arrive at this answer. This "answer-derived strategy" is then paired with the problem $x$ to form high-quality training samples.

**During Inference:** Since reference solutions are unavailable, we adopt a multi-candidate selection approach. In planning stage, we require LLM generates multiple potential plans and selects the most optimal one to proceed. The number of plan generations varies based on problem difficulty.

### B.3. Cost-Performance Analysis

In the main text, we restricted LPW to a single iteration to ensure a fair comparison; however, this constraint does not fully reflect the method's optimal performance. Therefore, we conduct an additional comparison using the 12-iteration setting specified in the original LPW paper. We evaluate both model performance and token consumption using Qwen2.5-Coder-7B as the backbone.

*Table 8.* Pass@1 accuracy comparison on code generation benchmarks using Qwen2.5-Coder-7B as the backbone. Benchmarks are categorized by difficulty levels. Note that PADA achieves superior performance on complex reasoning tasks (e.g., APPS-C and LiveCodeBench) compared to the computation-heavy LPW@12, demonstrating its superior reasoning capability in handling complex problems.

| Model | Method | easy | | medium | | difficult | | | Average |
|---|---|---|---|---|---|---|---|---|---|
| | | HumanEval | MBPP | MBPP+ | APPS-I | APPS-V | APPS-C | LCB | |
| | LPW@1 | 93.3 | 86.4 | 81.2 | 67.9 | 51.6 | 38.7 | 50.6 | 67.1 |
| Qwen2.5-coder-7B | LPW@12 | **97.6** | 89.7 | **86.7** | 72.7 | 52.7 | 40.7 | 53 | 70.4 |
| | PADA | 95.7 | **91.1** | 85.2 | **76** | **61.3** | **46.7** | **59.1** | **73.6** |

As illustrated in table 8, while scaling LPW to 12 iterations yields substantial gains on simpler benchmarks (e.g., HumanEval), this strategy exhibits diminishing returns on complex reasoning tasks. Specifically, on the challenging APPS-C benchmark, LPW@12 achieves only a marginal improvement of 2.0% over LPW@1 (38.7% → 40.7%). In stark contrast, PADA

achieves a remarkable accuracy of 46.7% on the same benchmark, significantly outperforming LPW@12 and demonstrating superior robustness (Zhou et al., 2025a; Wang et al., 2025b; Tian et al., 2026) in complex problem-solving.

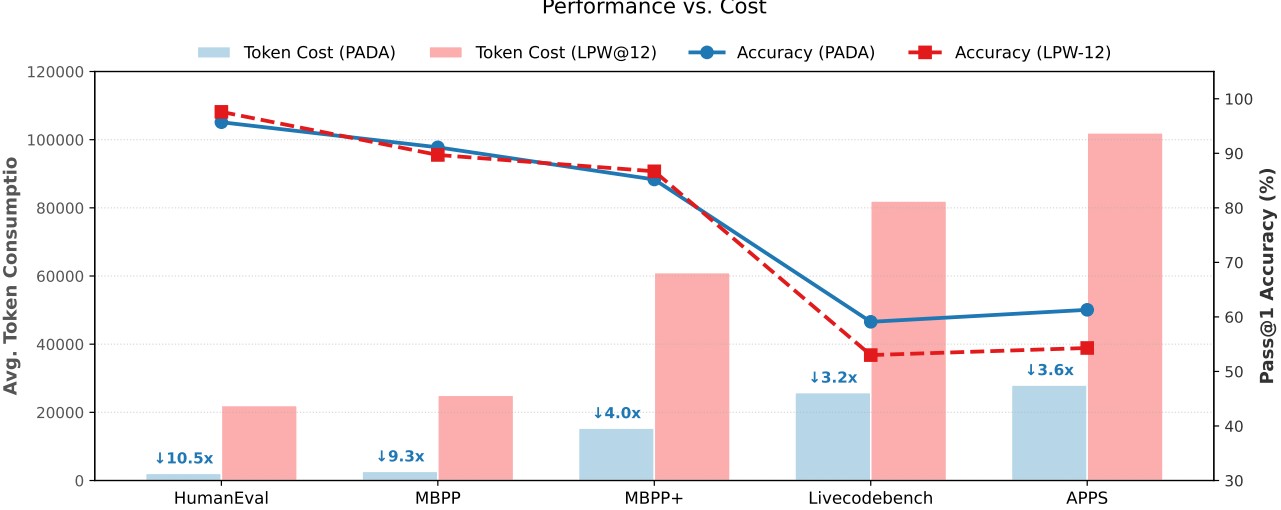

*Figure 13.* Analysis of the Performance-Efficiency Trade-off between PADA and LPW@12. The bars (left axis, lower is better) represent the average token consumption per problem, while the lines (right axis, higher is better) indicate Pass@1 accuracy. The comparison reveals that LPW@12 suffers from diminishing returns on complex benchmarks while incurring prohibitive computational costs (red bars). In contrast, PADA (blue) effectively breaks this bottleneck, achieving superior accuracy on challenging tasks like APPS with 3.6× less token consumption. On simpler tasks such as HumanEval, PADA reduces inference overhead by over 10×

As shown in Figure 13, LPW@12 incurs prohibitive costs to achieve these limited gains, with an average consumption of approximately 102k tokens per problem on APPS. Conversely, leveraged by its precise attention alignment mechanism, PADA attains superior performance with only 28k tokens, reducing inference costs by a factor of 3.6×. On the simpler HumanEval benchmark, this efficiency gap widens to over 10× (2.1k vs. 22k).

## B.4. The alignment of plan steps with code steps

To facilitate the sliding window mechanism, it is necessary to align the plan with the corresponding code segments. We present a representative case illustrating a problem description, its plan, and the generated code.

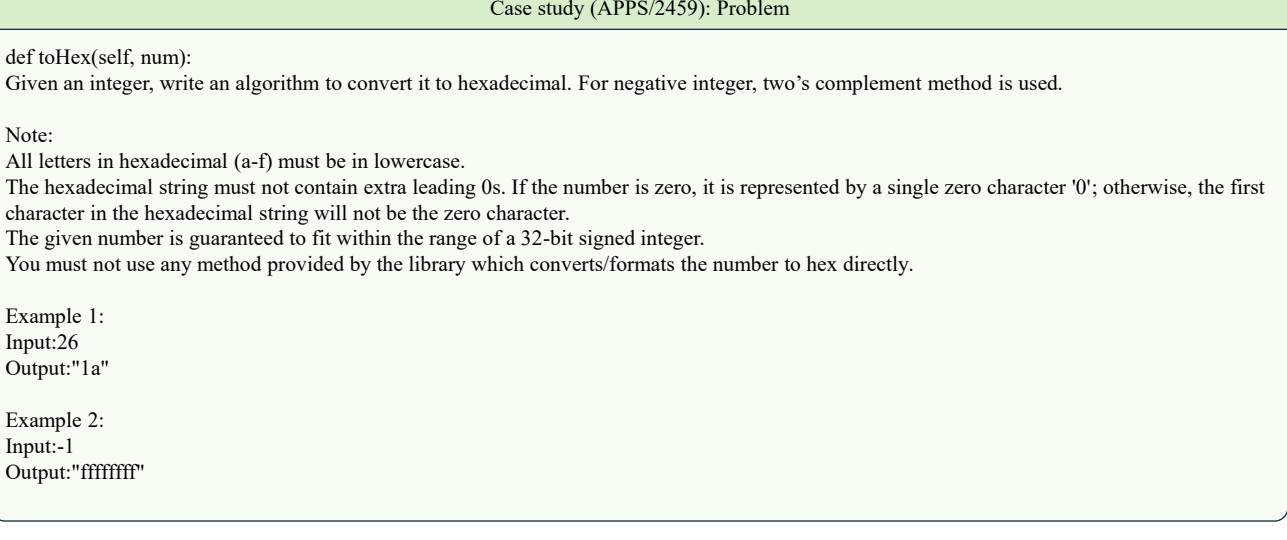

*Figure 14.* Problem description of APPS/2459

---

Case study (APPS/2459): Plan

[Algorithm] Base Conversion (Iterative Division/Modulo)

[STEP:1] Handle the special case where `num` is zero by immediately returning `\"0\"` to avoid leading zeros and extra processing.

[STEP:2] Initialize containers: a list to collect hexadecimal digits, a counter to cap output at 8 digits (32 bits), and a fixed mapping array for digits `0-9` and letters `a-f`.

[STEP:3] Iteratively compute the least-significant hex digit using `num % 16`, convert it via the mapping, and place it at the front of the result list to build the string in correct order.

[STEP:4] Update the number using integer division `num // 16`, increment the digit counter, and continue until either 8 digits are produced or the value becomes zero.

[STEP:5] Concatenate the collected digit characters into a final hexadecimal string without leading zeros and return it.

*Figure 15.* Plan of APPS/2459

---

Case study (APPS/2459): Code

```
[GEN_PLAN] (Current Plan: Handle the special case where num is zero and return \"0\".)
[GEN_CODE]
def toHex(self, num):
    if num == 0:
        return \"0\"

[GEN_PLAN] (Current Plan: Initialize digit list, counter, and hex digit mapping.)
[GEN_CODE]
    digits = []
    count = 0
    mapping = ['0', '1', '2', '3', '4', '5', '6', '7', '8', '9', 'a', 'b', 'c', 'd', 'e', 'f']

[GEN_PLAN] (Current Plan: Iteratively compute least-significant hex digit and place it at the front of digits.)
[GEN_CODE]
    while num != 0 and count < 8:
        digit = num % 16
        digits.insert(0, mapping[digit])

[GEN_PLAN] (Current Plan: Update num by integer division and increment the digit counter each iteration.)
[GEN_CODE]
        num //= 16
        count += 1

[GEN_PLAN] (Current Plan: Concatenate collected digits into the final hexadecimal string and return it.)
[GEN_CODE]
    result = \"\".join(digits)
    return result
```

*Figure 16.* Correct code of APPS/2459

As illustrated in the Figure 15 and Figure 16, during training sample generation, we explicitly instruct the teacher model to summarize the current plan prior to generating the corresponding code. We employ regular expressions to verify the alignment between plan steps and code segments. This ensures consistency, which is a prerequisite for the effective implementation of the subsequent sliding window attention alignment. Samples failing this verification are regenerated; if misalignment persists, they are discarded. Ultimately, this rigorous filtering process yielded 4,000 high-quality training samples. This efficiency-oriented observation is also consistent with a broader trend in efficient learning and distillation (Ma et al., 2026; Li et al., 2026b), where recent studies reduce training or supervision cost from different perspectives.

# C. Mathematical Formulation and Analysis of DID

## C.1. Assumptions and Derivation

### C.1.1. PROBLEM FORMULATION AND NOTATION

We model the Distilled Information Density (DID) optimization problem within the context of a Large Language Model (LLM) constrained by a fixed context window.

- **Context Constraint:** Let the total context length be a constant scalar $L \in \mathbb{N}^+$.

- **key token Set:** Let $S_K$ be the subset of selected tokens (key tokens) with cardinality $N = |S_K|$.

- **Density Parameter:** We define the key token density $\rho = \frac{N}{L}$, where $\rho \in (0, 1]$.

- **Macroscopic Order Parameter:** Let $S(\rho, t)$ denote the *Total Attention Scores* allocated to the set $S_K$ at training step $t$:

$$S(\rho, t) = \sum_{i \in S_K} A_i(t), \tag{17}$$

  where $A_i(t)$ is the softmax-normalized attention scores for token $i$. Note that $S(\rho, t) \in [0, 1]$.

**Objective:** We seek to maximize a utility function $DID(\rho)$ and prove the existence of a unique optimal density $\rho^* \in (0, 1)$.

### C.1.2. PROPOSITION I: THE SPATIAL INDUCTIVE BIAS (GAUSSIAN PRIOR IN PLAN-THEN-CODE)

**Proposition C.1.** *Recent studies have shown that although attention outputs in Large Language Models exhibit non-Gaussian behavior in the infinite-width limit—particularly under finite head counts and standard scaling—they can nonetheless be approximated as a Gaussian distribution conditioned on random similarity scores. This approximation holds in the regime of large network scale and numerous heads, especially when focusing on regions with high attention scores (Sakai et al., 2025; Noci et al., 2023). Accordingly, although generic code generation often entails long-range dependencies that might deviate from this prior, the "Plan-then-Code" paradigm explicitly imposes a structural locality bias. As evidenced in our Preliminary Experiment (Appendix A), the model's attention specifically over plan tokens exhibits strong concentration around the current reasoning step. This convergence of theoretical asymptotic behavior and our task-specific empirical observation justifies modeling the initial attention mass $S_0(\rho)$ as a centered Gaussian distribution $\mathcal{N}(\mu, \sigma^2)$, allowing us to rigorously quantify the attention coverage using the Error Function (erf).*

*Proof.* Based on the previous experiment, the model will allocate most attention to key tokens in the planning part under initial conditions. Therefore, we assume the "Top-K" plan tokens selection strategy selects tokens from the region of highest initial attention, the initial Total Attention Mass $S_0(\rho)$ is the integral of the Gaussian density over the selected interval $\left[-\frac{N}{2}, \frac{N}{2}\right]$:

$$S_0(\rho) = S(\rho, 0) = \int_{-\frac{\rho L}{2}}^{\frac{\rho L}{2}} \frac{1}{\sqrt{2\pi}\sigma} e^{-\frac{x^2}{2\sigma^2}} \, dx. \tag{18}$$

Letting $\kappa = \frac{1}{2\sqrt{2}\sigma}$ be a shape parameter derived from the attention head's focus width, this parameter simplifies to the Error Function:

$$S_0(\rho) = \text{erf}(\kappa \cdot \rho L). \tag{19}$$

Physically, this formulation encapsulates the intrinsic trade-off between coverage and efficiency. As the key token density $\rho$ approaches unity, the cumulative attention mass $S_0$ exhibits **saturation**, reflecting the finite attention budget. Concurrently, the Gaussian decay imposes a law of **diminishing returns**, where adding key tokens beyond the central "focal point" (the high-probability region) yields progressively smaller marginal gains in captured attention mass. $\square$

C.1.3. PROPOSITION II: IMPEDED MEAN-FIELD DYNAMICS (GRADIENT DILUTION)

**Proposition C.2.** *The temporal evolution of the attention mass $S(\rho, t)$ follows a logistic growth trajectory impeded by a dilution factor $(\rho L)^{-\beta}$, where $\beta \geq 1$.*

The differential equation governing the system is:

$$\frac{dS}{dt} = \eta \cdot \underbrace{\frac{1}{(\rho L)^{\beta}}}_{\text{Dilution Term}} \cdot \underbrace{S(1 - S)}_{\text{Geometric Term}} ,]. \tag{20}$$

Solving the differential equation yields the time-dependent state:

$$S(\rho, t) = \frac{1}{1 + Q(\rho) \cdot E(\rho, t)}, \tag{21}$$

where:

- **Initial Potential:** $Q(\rho) = \frac{1}{S_0(\rho)} - 1 = \frac{1}{\text{erf}(\kappa \rho L)} - 1$

- **Damping Factor:** $E(\rho, t) = \exp\left(-\frac{\eta t}{(\rho L)^{\beta}}\right)$

C.1.4. PROPOSITION III: THE DISTILLED INFORMATION DENSITY (DID) MODEL

**Definition C.3.** We define $DID(\rho)$ as the product of the **Information Capacity** and the **Average Attention Efficiency**.

$$DID(\rho) = \underbrace{\mathcal{I}(\rho)}_{\text{Capacity}} \cdot \underbrace{\bar{a}(\rho, t)}_{\text{Efficiency}} . \tag{22}$$

**Derivation:**

1. **Information Capacity($\mathcal{I}(\rho)$):** In code generation, tokens exhibit vast discrepancies in information density: substantive tokens (e.g., identifiers, keywords) carry high entropy, whereas syntactic tokens (e.g., separators, whitespace) contribute minimal semantic value.

   Since our method selects key tokens based on attention scores which correlate with semantic importance(Li et al., 2025b), this selection process implicitly sorts tokens by their information value. Consequently, the accumulation of information follows the **Law of Diminishing Marginal Utility**(Cover & Thomas, 2006): the initial high-ranking tokens resolve the primary logical uncertainty, while subsequent tokens (often syntax sugar or formatting characters) yield progressively smaller information gains. Mathematically, modeling the marginal gain of the $k$-th token as inversely proportional to its rank ($\propto 1/k$, following Zipf's Law(Zipf, 1949)), the cumulative information capacity scales logarithmically:

   $$\mathcal{I}(\rho) = \int_1^{\rho L} \frac{\alpha}{x} dx = \alpha \cdot \ln(\rho L). \tag{23}$$

2. **Attention Efficiency:** Defined as the Total Attention Mass normalized by the number of targets (Mean Field Intensity):

   $$\bar{a}(\rho, t) = \frac{S(\rho, t)}{\rho L}. \tag{24}$$

**Final Analytic Formula:**

$$DID(\rho) = \alpha \cdot \frac{\ln(\rho L)}{\rho L} \cdot \left[ \frac{1}{1 + \left(\frac{1}{\text{erf}(\kappa \rho L)} - 1\right) \cdot \exp\left(-\frac{\eta t}{(\rho L)^{\beta}}\right)} \right] . \tag{25}$$

C.1.5. (INFORMATION DENSITY & PERFORMANCE.

Recent studies suggest that higher information density in the context window significantly correlates with improved reasoning performance in LLMs (Li et al., 2025a; Heineman et al., 2025). DID serves as a quantitative metric for this post-distillation density.

Accordingly, we posit that the performance gain $\Delta$Pass@1 exhibits a **positive monotonic dependence** on DID, provided that the essential semantic logic remains intact. We model this relationship as:

$$\Delta\mathcal{P} \approx \mathcal{F}\big(\text{DID}(\rho)\big) \cdot \mathbb{1}\big(\mathcal{I}(\rho) \geq \mathcal{I}_{critical}\big), \tag{26}$$

where:

- $\mathcal{F}(\cdot)$ is a strictly increasing function ($\mathcal{F}' > 0$), representing the positive correlation derived from noise reduction and density maximization.

- $\mathbb{1}(\cdot)$ is the semantic completeness indicator, ensuring the hypothesis holds only when the minimal logical information $\mathcal{I}_{critical}$ is preserved.

The theoretical framework rigorously elucidates the **inverted-U trajectory** of Pass@1 performance observed in our experiments. Validated by this theory, we quantify the relationship between density and performance by fitting a regression curve to the multi-trial Pass@1 data against calculated DID values. Solving for the extremum of this fitted function allows us to mathematically pinpoint the **Optimal DID Interval** .

**C.2. The Analysis of $\beta \geq 1$**

C.2.1. THE ASYMPTOTIC BOUND (MATHEMATICAL PROOF FOR $\beta \approx 1$)

First, we establish that the asymptotic limit of attention allocation is strictly bounded by the inverse of the key token count, regardless of the target loss.

**Lemma C.4** (The Physical Limit of Attention). *Although the loss function targets $A_i \to 1$ for each key token, under the Softmax normalization constraint, the global optimal solution inevitably converges to $A_i \to \frac{1}{N}$.*

*Proof.* Consider the constrained optimization problem where we minimize the L2 distance between the attention distribution and the target:

$$\text{minimize} \quad f(\mathbf{A}) = \sum_{i=1}^{N}(A_i - 1)^2,$$

$$\text{subject to} \quad \sum_{i=1}^{N} A_i = 1, \quad A_i \geq 0.$$

Construct the Lagrangian function with multiplier $\lambda$:

$$\mathcal{L}(\mathbf{A}, \lambda) = \sum_{i=1}^{N}(A_i - 1)^2 + \lambda\left(\sum_{i=1}^{N} A_i - 1\right). \tag{27}$$

Taking the partial derivative with respect to $A_i$ and setting it to 0:

$$\frac{\partial \mathcal{L}}{\partial A_i} = 2(A_i - 1) + \lambda = 0 \implies A_i = 1 - \frac{\lambda}{2}. \tag{28}$$

Since $A_i$ depends only on $\lambda$ (and not on $i$), all $A_i$ must be identical. Substituting into the constraint $\sum A_i = 1$:

$$\sum_{i=1}^{N} A_i = N \cdot A_i = 1 \implies A_i^* = \frac{1}{N}. \tag{29}$$

**Conclusion:** The asymptotic limit is $A_\infty = \frac{1}{N}$. This implies that the gradient driving force per token scales intrinsically with $1/N$, thereby justifying a baseline dilution exponent of $\beta \approx 1$. $\square$

C.2.2. THE CHANNEL CAPACITY CONSTRAINT (THE ARGUMENT FOR $\beta > 1$)

From an engineering perspective, the Attention Mechanism has a fixed rank $d_{head}$.

- **Interference:** When $N$ approaches the effective rank of the attention subspace, linear independence is lost.

- **Super-linear Cost:** The "Search Cost" or interference noise increases super-linearly due to the Curse of Dimensionality in the key-query dot product space. This justifies a stronger penalty $\beta > 1$ (e.g., $\beta = 1.5$), representing the increasing difficulty of distinguishing signal from noise.

### C.3. Existence and Uniqueness of the Maximum DID($\rho^*$)

We employed Numerical Differentiation (Central Difference Method) to compute the first and second derivatives, ensuring mathematical rigor in identifying the stationary points. We rigorously verified the existence and uniqueness of the optimal density $\rho^*$ strictly within the physically feasible domain $\rho \in (0, 1]$.

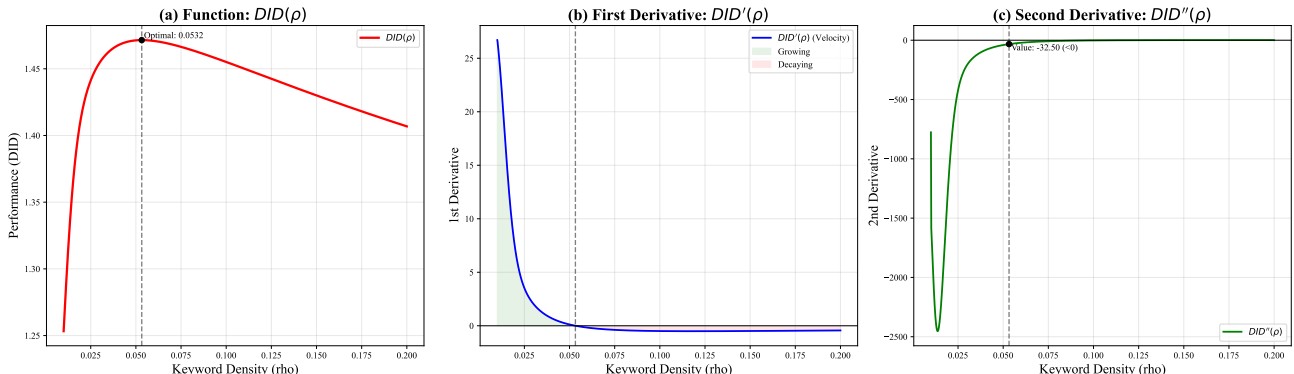

*Figure 17.* Theoretical Analysis of Distilled Information Density (DID). (a) The DID($\rho$) performance curve exhibits a unimodal trajectory. (b) The first derivative DID'($\rho$) crosses the zero axis exactly once, pinpointing the unique stationary point. (c) The second derivative DID"($\rho$) is negative at this intersection, mathematically confirming the existence of a global maximum and a robust optimal density interval within the physically feasible domain $\rho \in (0, 1]$.

Figure 17 demonstrates the presence of an optimal interval within the physically feasible domain $\rho \in (0, 1]$.

- Performance Curve ($DID(\rho)$): The curve rises steeply in the low-density region (information accumulation) and declines gradually in the high-density region (attention dilution), indicating a clear maximum.

- First Derivative ($DID'(\rho)$): The curve crosses the zero axis exactly once, transitioning from positive to negative. This zero-crossing mathematically confirms the existence and uniqueness of a stationary point (the peak).

- Second Derivative ($DID''(\rho)$): At the point where the first derivative is zero, the second derivative is significantly negative. This confirms that the stationary point is a global maximum, not a minimum or inflection point.

**Conclusion:**   The graphical analysis demonstrates that key token density optimization has a unique maximum within our defined domain. Although The unique peak $\rho^*$ represents the theoretical optimum within the domain $\rho \in (0, 1]$ through mathematical reasoning, in the Engineering Reality, The flatness of the curve around the peak defines a robust "Optimal Interval". Within this range, the model balances sufficient information intake with efficient attention allocation.

### C.4. The select of $k$, $\eta$ and $\tau$ based on DID

Empirical calibration across diverse base models reveals slight variation in the optimal density interval: Llama-3.2-3B-Instruct stabilizes at $[\rho_{min}, \rho_{max}] = [[0.05, 0.06]$, Qwen3-4B at $[0.045, 0.055]$, and Qwen2.5-Coder-7B-Instruct at $[0.04, 0.05]$. Based on this, we conservatively set the selection ratio $k$ to the maximum bound, $6\%$, when initializing the candidate pool. To adaptively identify key tokens, we introduce two dynamic thresholds, $\tau$ and $\eta$, grounded in the statistical distribution of token importance. The complete procedure of our proposed strategy is summarized in Algorithm 1.

**Logic Significance Threshold ($\tau$):** This threshold distinguishes between "perturbation noise" and "critical logic nodes." We observe that masking truly critical tokens results in a sharp spike in perplexity change ($\Delta$PPL), whereas masking irrelevant tokens causes only minor fluctuations. Thus, we perform perturbation analysis on the Disagreement Set ($\mathcal{D}$) and set $\tau$ at the most significant value gap (spike) in the ranked $\Delta$PPL distribution.

**Attention Focus Threshold ($\eta$):** This threshold differentiates "primary attention focus" from "background attention" within the Consensus Set ($\mathcal{C}$). Since teacher and student models may both attend to a broad range of tokens, we analyze the original Aggregated attention scores of tokens in $\mathcal{C}$. If a distinct gap exists between high-scoring and low-scoring subgroups, $\eta$ is set to this gap value to retain only the most attentive consensus tokens. Notably, any token selected from the Consensus Set via $\eta$ must still undergo perturbation verification against $\tau$ to ensure it contributes significantly to the generation logic.

---

**Algorithm 1** The dynamic select of $k$, $\eta$ and $\tau$ based on DID

---

**Require:** Student & Teacher Importance Vectors $\mathbf{v}_S, \mathbf{v}_T$, Context Length $L$, Selection Ratio $k$, Target Density Range $[\rho_{min}, \rho_{max}]$.
**Ensure:** Final Key Token Set $\mathcal{K}_{final}$.

1: **Phase 1: Initialization**
2: $N_{init} \leftarrow \lfloor L \times k \rfloor$;   $N_{min}, N_{max} \leftarrow \lfloor L \times \rho_{min} \rfloor, \lfloor L \times \rho_{max} \rfloor$
3: $\mathcal{K}_S \leftarrow \text{TopK}(\mathbf{v}_S, N_{init})$;   $\mathcal{K}_T \leftarrow \text{TopK}(\mathbf{v}_T, N_{init})$
4: $\mathcal{C} \leftarrow \mathcal{K}_S \cap \mathcal{K}_T$               ▷ Consensus Set
5: $\mathcal{D} \leftarrow (\mathcal{K}_S \cup \mathcal{K}_T) \setminus \mathcal{C}$           ▷ Disagreement Set
6: **Phase 2: Disagreement Verification (Base Logic)**
7: $\mathbf{S}_{\mathcal{D}} \leftarrow \text{PerturbationAnalysis}(\mathcal{D})$
8: $\tau \leftarrow \text{DetectValueSpike}(\mathbf{S}_{\mathcal{D}})$           ▷ Init Logic Threshold
9: $\mathcal{D}_{sig} \leftarrow \{t \in \mathcal{D} \mid \mathbf{S}_{\mathcal{D}}[t] > \tau\}$
10: $\mathcal{D}_{remain} \leftarrow \mathcal{D} \setminus \mathcal{D}_{sig}$       ▷ Store rejected tokens for potential fill
11: **Phase 3: Iterative Density Adaptation**
12: $\eta \leftarrow \text{DetectValueSpike}(\mathbf{v}_S[\mathcal{C}] \cup \mathbf{v}_T[\mathcal{C}])$      ▷ Init Attn Threshold
13: **loop**
14:      $N_{curr} \leftarrow |\mathcal{C}| + |\mathcal{D}_{sig}|$
15:      **if** $N_{curr} > N_{max}$ **then**          ▷ **Case A: Overflow**
16:          // Trust High-Attn $\mathcal{C}$, Verify Low-Attn $\mathcal{C}$ with $\tau$
17:          $\mathcal{C}_{low} \leftarrow \{t \in \mathcal{C} \mid \text{AvgAttn}(t) < \eta\}$
18:          $\mathbf{S}_{\mathcal{C}} \leftarrow \text{PerturbationAnalysis}(\mathcal{C}_{low})$
19:          $\mathcal{C}_{verified} \leftarrow \{t \in \mathcal{C}_{low} \mid \mathbf{S}_{\mathcal{C}}[t] > \tau\}$
20:          $\mathcal{K}_{final} \leftarrow (\mathcal{C} \setminus \mathcal{C}_{low}) \cup \mathcal{C}_{verified} \cup \mathcal{D}_{sig}$
21:          **if** $|\mathcal{K}_{final}| > N_{max}$ **then**
22:              $\eta \leftarrow \text{NextStricterSpike}(\mathbf{v}[\mathcal{C}], \eta)$      ▷ Raise bar for "High Attn"
23:              **continue**          ▷ Re-evaluate with stricter $\eta$
24:          **end if**
25:      **else if** $N_{curr} < N_{min}$ **then**         ▷ **Case B: Insufficient**
26:          // Fill from rejected Disagreement tokens
27:          $N_{needed} \leftarrow N_{min} - |\mathcal{C}|$
28:          $\mathcal{D}_{fill} \leftarrow \text{SelectTopN}(\mathcal{D}_{remain}, \mathbf{S}_{\mathcal{D}}, N_{needed})$
29:          $\mathcal{K}_{final} \leftarrow \mathcal{C} \cup \mathcal{D}_{sig} \cup \mathcal{D}_{fill}$
30:          **break**
31:      **else**
32:          $\mathcal{K}_{final} \leftarrow \mathcal{C} \cup \mathcal{D}_{sig}$
33:          **break**
34:      **end if**
35:      **break**          ▷ Safety break
36: **end loop**
37: **return** $\mathcal{K}_{final}$

---

# D. Attention Pattern Visualization Based on Causal Analysis

### D.1. Attention Analysis on Key Tokens

To address the problem that the attention patterns visualized in Figure 4 might stem from circular reasoning (i.e., the model merely overfitting to its own training targets rather than attending to semantically meaningful constraints), we conducted an independent verification experiment using a causal analysis framework adapted from LeaF (Guo et al., 2025). We established a "Causal Ground Truth" independent of our training pipeline. Following the causal analysis in LeaF, we calculated the causal contribution of each input token to the final code generation accuracy on a subset of 50 randomly sampled tasks from APPS and CodeContests (Li et al., 2022). For each task, we identified the set of tokens with the highest causal impact and randomly sampled $K = 6$ tokens to form a Ground-Truth Causal Set ($S_{gt}$). This randomness ensures we evaluate the model's robustness (Zhou et al., 2024) in capturing various types of constraints rather than a specific subset. we visualize the average attention scores allocated to $S_{gt}$ across all layers and heads during inference.

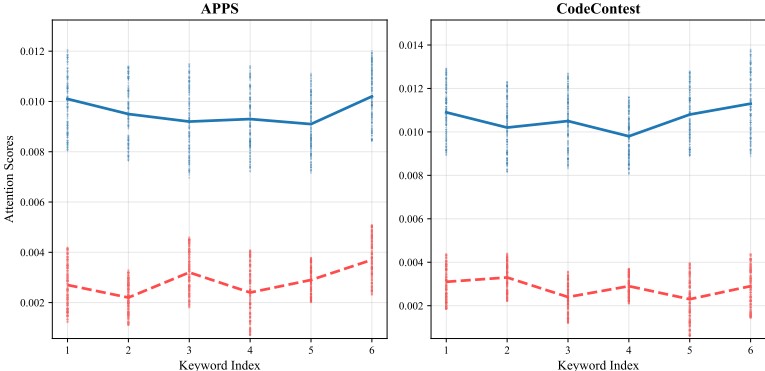

*Figure 18.* The attention scores allocated to the verified causal tokens. PADA consistently allocates more attention scores then base model.

As shown in Figure 19, PADA demonstrates a significantly higher overlap with $S_{gt}$ compared to the Base model. This aligns our attention mechanism with objective causality: PADA does not just follow its own predictions but effectively localizes the input spans that are causally necessary for correct program synthesis, as verified by the rigorous LeaF perturbation analysis.

### D.2. Attention Analysis on Non-Key Tokens

To rigorously rule out the hypothesis that our method simply flattens the attention distribution (i.e., indiscriminately increasing attention entropy across all tokens), we introduced a negative control group into our validation framework.Control Setup. In addition to the ground-truth causal set ($S_{gt}$) identified via the LeaF methodology, we randomly sampled a disjoint set of 6 non-causal tokens ($S_{noise}$) from the same context window for each instance.

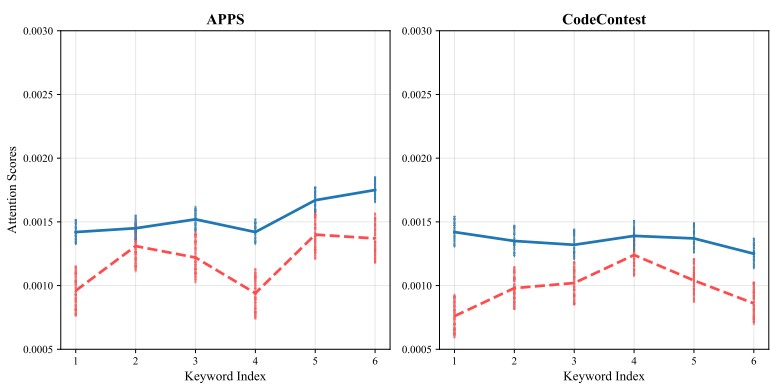

*Figure 19.* The attention scores allocated to $S_{noise}$.

The experimental results demonstrate that while PADA leads to a slight increase in attention scores for non-key tokens, the growth magnitude for key tokens is significantly larger. This substantial difference in growth rates indicates that PADA

effectively widens the gap between critical constraints and irrelevant context. By allocating a much larger proportion of the attention budget to key tokens compared to non-key tokens, our method successfully suppresses attention dispersion and ensures the model focuses on the most critical information.

## E. Motivation and Ablation of the Difficulty-Aware Gating

The purpose of the difficulty-aware gating mechanism is not to boost accuracy, but rather to ensure code conciseness (Li et al., 2025c). To isolate the gate's contribution, We evaluate PADA-coder (Qwen3-4B) with and without the gate on both simple (HumanEval) and complex (APPS) benchmarks:

*Table 9.* Ablation of Difficulty-Aware Gating on Accuracy and Conciseness

| Variant | HumanEval | | APPS | |
|---|---|---|---|---|
| | Pass@1 | Avg. Tokens | Pass@1 | Avg. Tokens |
| PADA-coder | 96.8% | 249 | 65.1% | 479 |
| w/o difficulty gating | 96.3% | 307 | 65.1% | 487 |

As shown in Table 9, the gate reduces token consumption on simple tasks ( 19% reduction) and yields a slight decrease on complex tasks, while maintaining accuracy. This demonstrates that the gate reduces inference overhead without compromising accuracy.

## F. Why teachers with similar scales perform poorly.

By conducting a teacher-student attention similarity analysis, we quantitatively demonstrate why teachers with similar parameter scales perform poorly.

We extract the attention distributions of the student model (Qwen2.5-coder-7B) and the respective teacher models on the APPS test set. We select the specific target layers and target KV heads for each model, exactly as detailed in Table 9 (Appendix I). We first average the attention weights across the selected attention heads within these target layers to obtain a macroscopic layer-level attention distribution. We then calculate the cross-model JSD for these specific layers and average the results. The results are presented in Table R1.

*Table 10.* Average JSD Distance Between Student and Teacher Models (APPS Test Set)

| Base Model | Teacher Model | JSD Distance |
|---|---|---|
| Qwen2.5-coder-7B | Qwen3-4B | 0.0724 |
| | Qwen3-32B | 0.3466 |

As demonstrated in Table R1, the JSD distance between the base model and the similar-scale teacher (Qwen3-4B) is small (0.0724), whereas the larger-scale teacher (Qwen3-32B) exhibits a higher distance (0.3466). This mathematically confirms that similar-scale models share highly overlapping attention patterns, limiting their effectiveness as teachers since they fail to provide distinct supervisory signals. In contrast, the dissimilar attention pattern of the larger model successfully supplements more crucial key tokens attention information. We will add this analysis to the revised version.

## G. Training Overhead Analysis

We report detailed runtime and memory analyses (Li et al., 2026a) of PADA compared with standard fine-tuning under identical hardware configurations.

**Attention Extraction.** Attention extraction in PADA is a one-time offline process on 8×NVIDIA A100 (80GB) GPUs, jointly extracting attention scores from the teacher and student models. Processing around 4K samples takes about 1.2 hours.

**perturbation analysis.** perturbation analysis in PADA is a one-time offline process on 8×NVIDIA A100 (80GB) GPUs. Processing 4K samples requires approximately 8 minutes. This step is parallelizable and executed only once.

**Training Overhead.** We further measure the end-to-end training overhead over 2 epochs on a NVIDIA A100 (80GB) GPU. PADA incurs an additional 12–15% training time compared to standard fine-tuning.

# H. Evaluation Benchmarks

### Code Generation Benchmarks

APPS categorizes coding problems into three difficulty tiers: Introductory(1000), Interview(3000), and Competition(1000). We selected 150, 450, and 150 validated samples from each difficulty level respectively. It evaluates models' adaptability across a wide spectrum of cognitive loads, requiring solutions ranging from simple syntax implementation to complex algorithmic reasoning.

HumanEval consists of 164 handwritten Python programming problems that require completing function bodies based on docstrings and unit tests. It serves as a standard benchmark to assess functional correctness and the model's ability to translate natural language specifications into executable code.

MBPP includes 427 entry-level Python problems designed to test fundamental programming concepts and standard library usage.

MBPP+ provides a rigorously sanitized version (399 samples) of MBPP with enhanced test consistency, ensuring a more reliable evaluation of the model's grasp of basic control flow and logic.

LiveCodeBench(v5) collects 164 programming contest problems published between September 2024 and February 2025. This subset is specifically curated to post-date the training cutoff of our latest released Qwen3-4B base model, evaluating generalization on out-of-distribution samples and strictly preventing test data contamination.

### Mathematical Reasoning Benchmarks

GSM8K features 1319 high-quality grade school math word problems that demand multi-step reasoning. It evaluates the model's capability to perform Chain-of-Thought (CoT) reasoning and decompose complex logical statements into sequential arithmetic operations.

MATH-500 represents a challenging subset (500 samples) of the MATH dataset, covering diverse advanced topics such as algebra, calculus, and geometry. It tests the model's proficiency in solving competition-level mathematical problems that require deep domain knowledge and heuristic problem-solving skills.

# I. Open-source Instruct Models

Below are download links for five open-source models:

**Qwen3-32B:** https://huggingface.co/Qwen/Qwen3-32B

**Llama-3.3-70B-Instruct:** https://huggingface.co/meta-llama/Llama-3.3-70B-Instruct

**Qwen3-4B:** https://huggingface.co/Qwen/Qwen3-4B

**Qwen2.5-coder-7B-Instruct:** https://huggingface.co/Qwen/Qwen3-4B/Qwen2.5-coder-7B-Instruct

**Llama-3.2-3B-Instruct:** https://huggingface.co/meta-llama/Llama-3.2-3B-Instruct

# J. Detailed Training Settings

Specifically, we impose attention alignment constraints on the final few transformer layers and on only half of the attention heads in LLM, while leaving the remaining attention heads unconstrained in order to preserve the model's intrinsic capacity for unconstrained text generation. The complete set of training hyper-parameters is listed in the table 11.

*Table 11.* Training hyper-parameters

| Model | Hyper-parameter | Value |
|-------|-----------------|-------|
| Qwen3-4B | Target Layers | 20-32 |
| | Target KV Heads | [0, 1, 2, 3] |
| Qwen2.5-coder-7B-Instruct | Target Layers | 20-28 |
| | Target KV Heads | [0, 1] |
| Llama-3.2-3B-Instruct | Target Layers | 20-28 |
| | Target KV Heads | [0, 1, 2, 3] |

## K. Limitations and Future work

Our work has the following limitations: (1) **Dependency on Plan Quality:** As PADA follows the "Plan-then-Code" paradigm, its performance is inherently constrained by the quality of the generated plans. For small-scale LLMs, producing high-quality plans is challenging, which may ultimately constrain the final code accuracy. Future work could explore joint distillation of both planning logic and attention alignment to enable the model to generate higher-quality plans independently. (2) **The lack of programming Language Diversity:** Although we get much improvement on Python-based benchmarks, the effectiveness of PADA has not been extensively tested on other languages like C, Rust, or Assembly, which involve different syntactic and logical constraints. Extending PADA to a broader range of programming languages remains an important direction for future research.

