# OpenReview forum: "PADA-Coder: Improving Plan-Following Code Generation via Perturbation-Verified Attention Distillation and Dynamic Alignment"
_ICML.cc/2026/Conference — ICML 2026 regular_

### Official Review · Reviewer_vXsy · 2026-02-25

**Soundness:** 3
**Presentation:** 3
**Significance:** 2
**Originality:** 2
**Overall Recommendation:** 4
**Confidence:** 2

**Summary:**

This paper addresses the attention allocation imbalance issue in the Plan-then-Code paradigm by proposing the PADA framework. Core innovations include:
Identifying two attention problems in small-scale LLMs (<8B) during Plan-then-Code:
1) Attention Drift: As generation progresses, attention becomes captured by recently generated code, neglecting the initial plan
2) Attention Dispersion: Attention spreads evenly across irrelevant tokens, failing to focus on critical constraints.

They employ:
1) Distillation Information Density (DID): Quantifies token importance through perturbation analysis to construct a maximum information density attention target matrix.
2) Dynamic Attention Alignment: Dynamically adjusts attention alignment strength by combining a sliding window mechanism with difficulty-aware gating.

Across 7 benchmarks (HumanEval, MBPP, MBPP+, APPS-I/V/C, LiveCodeBench), using 2 teacher models (Qwen3-32B, Llama3.3-70B) and 3 student models (Qwen3-4B, Qwen2.5-coder-7B, Llama3.2-3B) achieved improvements of up to 16.7%.

**Compliance With Llm Reviewing Policy:**

Affirmed.

**Final Justification:**

The author has resolved some of my issues.

**Key Questions For Authors:**

See Weaknesses.

**Limitations:**

Yes.

**Strengths And Weaknesses:**

> Strengths
1.  Attention visualization (Figures 1, 4, 5, 6–10) provides an intuitive demonstration of the phenomena of attention drift and dispersion.
2. The Perturbation-Verified mechanism not only relies on teacher models but also validates token importance through ∆PPL (perplexity increment), mitigating teacher imitation bias.
3. From an information-theoretic perspective (Gaussian prior, Zipf's law, channel capacity), we rigorously derive the optimal token density (4%-6%) and prove its uniqueness (Appendix C).

> Weaknesses
1. The ablation study is inconclusive and cannot be directly mapped to existing method modules.
2. The method lacks sufficient detail in its description. Although Figure 2 illustrates the concept, it lacks technical details on how to achieve alignment between plan steps and code segments.
3. Writing and Expression Improvement. Refine the symbol system and provide a symbol reference table.
4. Add teacher-student attention similarity analysis (e.g., JSD distance) to explain why teachers with similar scales perform poorly.
5. The current DID range [0.04, 0.06] is an empirical value. We recommend exploring the relationship between task difficulty and optimal density (e.g., whether simpler tasks require lower density?).

---

> ### Author Rebuttal · Authors · 2026-03-31
>
> Thank you for recognizing our visualizations, perturbation mechanism, and theoretical derivation. We detail our responses to your concerns below.
>
> ***W1: The ablation study is inconclusive***
>
> Ablation studies for our primary core modules (the Sliding Window, Perturbation strategy, and Attention Alignment) are already provided in Table 4, Page 7 in our paper.
>
> To fully address your concern, we further ablate the difficulty-aware gating mechanism. The results demonstrate that while the Pass@1 remains almost unchanged, the token consumption on HumanEval is reduced by approximately 19%. This proves the mechanism's primary objective: ensuring code conciseness rather than boosting raw accuracy. For details, please refer to our response to Reviewer AnDX (W3). We will incorporate these results into the revised manuscript.
>
> ***W2: The method lacks sufficient detail***
>
> As detailed in Appendix B.4, we achieve strict alignment by instructing the teacher model to summarize the current plan prior to coding and verifying this alignment using regular expressions. To further demonstrate the robustness of this method, we have supplemented additional experiments (achieving a 96.3% success rate on the APPS-Competition dataset). For details, please refer to our response to Reviewer AnDX (W4).
>
> ***W3: Writing and Expression Improvement.***
>
> We thank the reviewer for this constructive suggestion. We recognize that providing full explanations of the mathematical symbols solely within the main text is insufficient for optimal readability. We will add a Symbol Reference Table to the revised version. This table will consolidate all mathematical symbols used in our methodology and integrate their corresponding explanations directly from the main text.
>
> ***W4: why teachers with similar scales perform poorly.***
>
> We thank the reviewer for this insightful suggestion. By conducting a teacher-student attention similarity analysis, we quantitatively demonstrate why teachers with similar parameter scales perform poorly.
>
> We extract the attention distributions of the student model (Qwen2.5-coder-7B) and the respective teacher models on the APPS test set. We select the specific target layers and target KV heads for each model, exactly as detailed in Table 9 (Appendix I). We first average the attention weights across the selected attention heads within these target layers to obtain a macroscopic layer-level attention distribution. We then calculate the cross-model JSD for these specific layers and average the results. The results are presented in Table R1.
>
> Table R1: Average JSD Distance Between Student and Teacher Models (APPS Test Set)
> |Base Model|Teacher Model|JSD Distance|
> |-|-|-|
> |Qwen2.5-coder-7B|Qwen3-4B|0.0724|
> ||Qwen3-32B|0.3466|
>
> As demonstrated in Table R1, the JSD distance between the base model and the similar-scale teacher (Qwen3-4B) is small (**0.0724**), whereas the larger-scale teacher (Qwen3-32B) exhibits a higher distance (**0.3466**). This mathematically confirms that similar-scale models share highly overlapping attention patterns, limiting their effectiveness as teachers since they fail to provide distinct supervisory signals. In contrast, the dissimilar attention pattern of the larger model successfully supplements more crucial key tokens attention information. We will add this analysis to the revised version.
>
> ***W5: relationship between task difficulty and optimal density.***
>
> To address your concern, we clarify that the optimal Distillation Information Density (DID) interval does not exhibit a significant dependence on task difficulty. The optimal DID is mathematically derived from the plan's information density and the statistical distribution of the model's attention. As task difficulty increases, the reasoning steps become more complex, resulting in correspondingly longer plans (please refer to Table R1 and table R3 for **Reviewer AnDX**). Consequently, the absolute number of key tokens increases. However, the **relative information density** of the plan remains stable across different difficulty levels. Therefore, the optimal density ratio does not fluctuate significantly.
>
> To empirically prove this stability, we conduct an additional experiment. We train the Llama-3.2-3B-Instruct on easier datasets (a combination of APPS-Introductory and MBPP, totaling 1,450 samples) and evaluate it on the HumanEval benchmark. By fitting the empirical data to our theoretical DID model (Formula 7), the derived optimal density stabilizes at approximately **0.05**. This value remains within our proposed optimal interval and is almost identical to the **0.051** optimal DID obtained when training on the APPS dataset. This experiment confirms that the optimal DID range is an inherent property of the plan's information structure and the attention mechanism, proving its robustness across varying levels of task difficulty. We will incorporate the supplementary experiment into the revised version.

---

> > ### Author Rebuttal · Reviewer_vXsy · 2026-04-02
> >
> > Thank you for your response.

---

> > > ### Author Response · Authors · 2026-04-02
> > >
> > > We thank you for your thorough review and insightful suggestions. Your comments have significantly helped us improve the paper by supplementing the ablation study for the gating mechanism, clarifying the method details and mathematical symbols, and adding robust quantitative analyses regarding both attention similarity (JSD) and the stability of optimal information density. We sincerely appreciate your constructive feedback, which has made our revised version more comprehensive, empirically solid, and clearer to readers.

---

### Official Review · Reviewer_AnDX · 2026-03-10

**Soundness:** 3
**Presentation:** 3
**Significance:** 3
**Originality:** 3
**Overall Recommendation:** 4
**Confidence:** 3

**Summary:**

This paper studies plan-adherence in LLM code generation, motivated by the observation that adding explicit plans can substantially increase context length and may make it harder for smaller models to focus on the most relevant parts of the prompt. The authors propose PADA, a training method that identifies keycontext tokens using teacher/student attention signals together with perturbation-based filtering, and then applies a dynamic attention-alignment objective to encourage the student to attend more strongly to those tokens during code generation. The method uses stepwise plan decomposition and aligns code segments with plan steps so that attention supervision can be applied locally over relevant plan windows. Empirically, the paper reports improved results over the base models and several alternative methods across multiple code-generation benchmarks, with additional smaller gains on math reasoning tasks.

**Compliance With Llm Reviewing Policy:**

Affirmed.

**Final Justification:**

I have increased my score to 4 after the author rebuttal. The paper has a useful core idea and strong empirical results in the setting studied, with consistent gains over the base models and compared baselines across the reported code-generation benchmarks. I also found the added value of the attention-alignment component reasonably supported by the ablations.

My main original concerns were about attribution and framing: in particular, whether the gains should be attributed to the proposed attention-alignment mechanism versus the broader answer-guided, plan-conditioned training setup, and whether the evidence fully supported the paper’s broader long-context / long-plan framing. The rebuttal addressed several of these concerns in a meaningful way. Most importantly, the added fixed-plan experiment strengthens the claim that the method improves plan following rather than only plan generation, and the new results on alignment robustness were also helpful.

I still think the evidence is stronger for moderately long composite prompts than for the broadest version of the long-context / long-plan framing. I also remain less convinced by the evidence for the necessity of the difficulty-aware gating mechanism, since the added ablation shows only modest effects, mainly on output length for simpler tasks. However, after considering the rebuttal, I no longer view these issues as outweighing the practical contribution of the paper. Overall, I view this as a promising empirical contribution with some remaining framing and method-justification limitations, but one that is above threshold.

**Key Questions For Authors:**

1. The paper is framed around improving plan adherence via perturbation-verified attention distillation/alignment, but both PADA and the “w/o attention alignment” variant appear to be trained on answer-guided, plan-conditioned supervision. Could the authors clarify what portion of the gains they believe comes from:
    (a) stronger plan generation from the data construction,
    (b) general SFT on these curated samples, and
    (c) the proposed attention-alignment mechanism itself?

2. Since training-time plans are derived using the reference solution, could the authors help disentangle whether PADA mainly improves the quality of generated plans, the ability to follow a fixed plan, or both?

3. Could the authors report the distribution of plan lengths, both in number of steps and token counts, for teacher and student generated plans across the main settings? The paper motivates the method using long context and longer plans, but the clearest analysis appears to be on short 5-step APPS plans, and the final mechanism is a 3-step sliding window.

4. Relatedly, how would this PADA interact with newer code-generation settings that include explicit reasoning / thinking tokens, which can substantially increase context length (e.g. Qwen3)?

**Limitations:**

Yes (limitations are in appendix J)

**Strengths And Weaknesses:**

**Strengths**

1. The core idea is interesting and practically relevant: identify a sparse set of important plan tokens and train the student to attend to them more reliably during code generation.

2. The empirical results are strong in the setting studied. PADA improves over the base models and the compared baselines across the reported code-generation benchmarks for all three student backbones, and the ablations suggest that the attention-alignment component provides additional value beyond the underlying fine-tuning setup.

3. The paper is generally well organized and the high-level method easy to follow. It also includes some targeted analyses beyond headline benchmark numbers, including a plan-sensitivity experiment using correct versus incorrect plans, which at least partially supports the claim that the method increases reliance on plan information.


**Weaknesses**

1. The main attribution story is not cleanly separated. The paper is framed primarily as improving plan adherence via perturbation-verified attention distillation/alignment, but the training recipe also includes answer-guided, plan-conditioned supervision. In particular, the “w/o attention alignment” ablation is described as standard fine-tuning and already yields a large fraction of the improvement over the base model, while the teacher-free variant also remains strong. As a result, it is currently hard to disentangle how much of the gain comes from better plan generation due to the curated training data, how much comes from general fine-tuning on those samples, and how much comes specifically from the proposed attention-alignment mechanism.

2. The framing emphasizes long-plan adherence more strongly than the evidence currently supports. The paper motivates the problem in terms of increasingly long plans and long-range dependencies, but the clearest attention analysis is based on what seem like relatively short 5-step APPS plans, and the final method uses a 3-step sliding window over adjacent plan steps. I did not see a quantitative characterization of plan lengths across the evaluation set or a length-stratified analysis showing that the gains are concentrated on genuinely long plans. As written, the evidence seems strongest for improving adherence to short-to-moderate local plan structure.

3. The role and need for the difficulty-aware gating mechanism are not fully motivated in the current presentation. The paper states that the gate is introduced to prevent “plan over-interpretation” on simple segments, but this failure mode is not clearly described, and there is no ablation isolating the gate’s contribution. As written, the component may be useful in practice, but it is not yet clear how central it is to the method or whether it is just empirically useful on the evaluated datasets.

4. A few implementation details that are central to reproducibility are under-specified. In particular, the plan–code alignment procedure seems to be very important to the method, but the paper only gives a high-level description in the appendix that alignment is produced through interleaved `[GEN_PLAN]` / `[GEN_CODE]` generation by the teacher and verified with regular expressions, with failed examples regenerated or discarded. It is also not clear how this procedure changes in the teacher-free setting, or how robust it is as plans and code outputs become longer and more structurally complex. These details do not invalidate the results, but they should be surfaced more clearly.

5. [Minor] The APPS evaluation uses a subset of the test set rather than the full benchmark, and this is easy to miss as it is only mentioned in the appendix. Using the subset is not a problem, but it should be made clear in the main body.

---

> ### Author Rebuttal · Authors · 2026-03-31
>
> Thank you for recognizing our core idea and empirical results. We address the issues you raised below.
>
> ***w1 & Q1, Q2: The main attribution***
>
> To clarify the exact source of performance gains, we break down our method's attribution:
>
> Regarding Q1(a), we clarify that our training recipe does not include any training for plan generation. PADA is strictly designed to improve plan following. To prove this, we conduct an evaluation where both the Base model and PADA-Coder were provided with the same correct plan. As shown in Table 3, Page 7 in our paper, PADA-Coder achieves a **+8.1%** average improvement over the Base model, which confirms PADA enhances plan following, not generation.
>
> Regarding Q1(b) general SFT vs. Q1(c) the attention-alignment mechanism: Long contexts often trigger structural collapses (early termination, format corruption) in small-scale LLMs. General SFT ("w/o attention alignment") improves the Base model's performance primarily by mitigating these formatting issues, but it fails to resolve Attention Allocation Imbalance. As shown in our ablation study (Table 4, page 7), while SFT alone improves the Base model to 57.8% on APPS, applying our proposed PADA mechanism delivers an additional **+7.3%** improvement (reaching 65.1%). This demonstrates that attention-alignment mechanism provides an independent gain.
>
> ***W2 & Q3, Q4: Framing of Long Plans***
>
> First, we would like to clarify that our method's core motivation addresses the attention allocation imbalance caused by the **overall long context**, rather than just plan length. In the standard "plan-then-code" paradigm, the input prompt is a composite of constraint, problem, plan, visible tests, and verification report. These components collectively form a long context. Furthermore, when reasoning models generate thinking traces, the context length increases even further. Because the plan occupies a relatively small fraction of this context, the model's attention is diluted, leading it to allocate attention to non-key tokens.
>
> Second, harder problems do generate longer plans. We sample 50 instances each for 5, 7, and 9-step plans from the APPS training set to illustrate the relationship between plan length and total context length.
>
> Table R1: Average Token Lengths for Different Plan Steps
> |Steps|Prompt Length|Plan Length|Plan Proportion|
> |-|-|-|-|
> |5-steps|2797|332|11.9%|
> |7-steps|3952|457|11.6%|
> |9-steps|4997|583|11.7%|
>
> As shown in Table R1, the plan consistently accounts for about 12% of the total prompt length. This dilution directly causes the model to neglect the plan during code generation, highlighting the critical need for our attention-alignment mechanism to maintain focus on the plan's key tokens.
>
> ***W3: Motivation and Ablation of the Difficulty-Aware Gating***
>
> The purpose of the difficulty-aware gating mechanism is not to boost accuracy, but rather to ensure code conciseness. To isolate the gate's contribution, We evaluate PADA-coder (Qwen3-4B) with and without the gate on both simple (HumanEval) and complex (APPS) benchmarks:
>
> Table R2: Ablation of Difficulty-Aware Gating on Accuracy and Conciseness
> |Variant|HumanEval(Pass@1)|HumanEval(Avg. Tokens)| APPS(Pass@1)|APPS(Avg. Tokens)|
> |-|-|-|-|-|
> |PADA-coder|96.8%|249|65.1%|479|
> |w/o difficulty gating|96.3%|307|65.1%|487|
>
> As shown in Table R2, the gate reduces token consumption on simple tasks (~19% reduction) and yields a slight decrease on complex tasks, while maintaining accuracy. This demonstrates that the gate reduces inference overhead without compromising accuracy.
>
> ***W4: Details of Plan-Code Alignment and Robustness***
>
> We have detailed the plan-code alignment procedure in Appendix B.4. Here, we provide further clarifications and additional empirical evidence:
>
> **1.Alignment in the Teacher-Free Setting:** The alignment procedure in the teacher-free setting is entirely consistent with the teacher model approach described in our paper.
>
> **2.Robustness Across Complexity and Length:** To quantify the robustness of this alignment procedure as plans become longer, we track the alignment success rate during the generation of our training data across different problem difficulty levels.
>
> Table R3: Alignment Success Rate Across Different Problem Complexities on APPS (Qwen3-32B as generator)
> |Problem Difficulty|Avg. Plan Steps|Alignment Success Rate (1st Attempt)|Final Yield Rate (up to 3 Retries)|
> |-|-|-|-|
> |Introductory|4.8|98.6%|99.7%|
> |Interview|6.7|96.4%|98.2%|
> |Competition|8.4|92.6%|96.3%|
>
> As shown in Table R3, our alignment procedure maintains high robustness, our maximum 3-retry mechanism ensure a final yield rate exceeding **96%** across all difficulty levels. This confirms that our alignment strategy does not suffer from significant degradation as complexity increases.
>
> ***W5***
>
> We will incorporate detailed information about the test dataset into the main text.
>
> We will incorporate the empirical results mentioned above into the revised version.

---

> > ### Author Rebuttal · Reviewer_AnDX · 2026-04-03
> >
> > Thank you for the detailed rebuttal. The added experiments addressed several of my main concerns, particularly around fixed-plan following and the role of the difficulty-aware gate, and increased my confidence in the paper’s practical contribution. I still think the evidence is stronger for moderately long composite prompts than for the broader long-context / long-plan framing, but overall I found the rebuttal sufficiently persuasive and have adjusted my score accordingly.

---

> > > ### Author Response · Authors · 2026-04-03
> > >
> > > Thank you for your careful review and insightful suggestions throughout the review process. Your comments have helped us significantly improve the paper by providing stronger evidence to disentangle plan generation from fixed-plan following, isolating the exact role of the difficulty-aware gate with new ablations, and clarifying the robustness of our plan-code alignment procedure. We sincerely appreciate your supportive assessment and your decision to adjust the score. We believe these revisions have significantly enhanced the clarity, rigor, and overall readability of our work.

---

### Official Review · Reviewer_yLPD · 2026-03-13

**Soundness:** 4
**Presentation:** 4
**Significance:** 4
**Originality:** 4
**Overall Recommendation:** 5
**Confidence:** 3

**Summary:**

The authors identify two primary problems in LLM code generation: "Attention Dispersion," where attention is spread too broadly across irrelevant tokens, and "Attention Drift," where the model's focus shifts to recently generated code rather than the guiding plan. Towards the problems, the authors propose Perturbation-Verified Attention Distillation and Dynamic Alignment (PADA). PADA extracts key tokens using perturbation analysis to maximize Distillation Information Density (DID) and employs a sliding window alongside a difficulty-aware gating mechanism to dynamically align the model's attention during training. The method is evaluated on models like Qwen and Llama across diverse benchmarks, demonstrating significant improvements.

**Compliance With Llm Reviewing Policy:**

Affirmed.

**Final Justification:**

The rebuttal addressed my main concerns.

**Key Questions For Authors:**

- Generalization on other languages. Given that different programming languages have drastically different semantic densities, how sensitive are the empirically calibrated DID density thresholds (e.g., 4%-6%) to languages other than Python? Would the perturbation threshold $\tau$ need manual recalibration per language?

**Limitations:**

yes

**Strengths And Weaknesses:**

Strengths:
- Novel approach. The conceptualization of Distillation Information Density (DID) to find an optimal trade-off between attention coverage and efficiency is theoretically rigorous and well-motivated. Furthermore, utilizing perturbation analysis (measuring Perplexity Increment) to filter out "Teacher Imitation Bias" is novel.
- Good empirical results. PADA delivers impressive performance gains, improving Pass@1 by up to 16.7% and consistently outperforming state-of-the-art baselines like SPA, LPW, and LeaF across multiple datasets. Notably, it enables small-scale models to achieve performance comparable to ultra-large LLMs like Gemini-2.5-pro on challenging benchmarks like APPS.
- Cost-efficiency. The authors demonstrate that PADA achieves superior accuracy on complex reasoning tasks compared to heavily iterative methods while reducing inference token consumption.

Weaknesses:
- Pipeline complexity and overhead. The training data construction is highly complex. The offline attention extraction and perturbation analysis steps are expensive.
- Dependency on plan quality. The effectiveness of PADA heavily depends on the quality of the initial plan. If the provided plan is incorrect, the forced attention alignment could actually reinforce wrong reasoning, leading to performance degradation.

---

> ### Author Rebuttal · Authors · 2026-03-31
>
> Thank you for appreciating the novelty, empirical results, and cost-efficiency of PADA. We address the issues you raised below.
>
> ***W1: Pipeline complexity and overhead.***
>
> To address concerns regarding the computational overhead, we provide a direct offline cost comparison with LeaF. Building upon the runtime analysis presented in Appendix F, Table R1 illustrates that while PADA introduces an offline processing pipeline, its overall computational cost remains highly competitive and requires significantly fewer resources than similar distillation methods.
>
> Table R1: Offline Processing Overhead Comparison
> | Phase | Metric | PADA (Ours) | LeaF |
> | - | - | - | - |
> | Offline | Process 1 (GPUs) | Attention Extraction (8×A100) | Gradient Computation (8×A100) |
> | | Process 1 Time | ~1.2 hours (4K samples) | ~3 hours (7K samples) |
> | | Process 2 (GPUs) | Perturbation Analysis (8×A100) | Counterfactual Gen (8×A100) |
> | | Process 2 Time | ~8 minutes (4K samples) | 25 mins - 1.43 hours (26K samples) |
>
> In summary, both methods require one-time offline preprocessing. While PADA involves multiple extraction and analysis steps, it completes this pipeline in slightly less time overall compared to LeaF's requirements. This demonstrates that the overhead introduced by our data construction strategy is an acceptable trade-off for the resulting performance improvements.
>
> ***W2: Dependency on Plan Quality***
>
> As we detailed in Appendix B.1, the quality of the initial plan indeed has a slight impact on the model's performance. However, we address this practical concern from two empirical perspectives:
>
> (1)Small-parameter models already demonstrate a highly reliable capacity for generating correct plans. Even on the most challenging APPS benchmark, the plan generation accuracy reaches **92.3%** (For detailed accuracy, please refer to Table R2 for **Reviewer 84rV**). Consequently, for the vast majority of coding tasks, the risk of reinforcing incorrect reasoning through attention alignment is minimal.
>
> (2)If a model consistently fails to generate a correct plan after multiple iterations, it strongly indicates that the problem exceeds the model's intrinsic reasoning capabilities.
>
> To empirically validate this, we isolate 58 problems from the APPS dataset where the model fail to generate a correct plan. We tested these specific, challenging problems using other baseline methods, and all attempts yield exceptionally poor results(LPW@12 of 5.2%, LeaF of 3.4%, For detailed analysis, please refer to the response for W3 & Q1 to **Reviewer 84rV**).
>
> Therefore, the theoretical degradation caused by incorrect plans rarely manifests as a primary failure mode. For the minority of tasks where plan generation fails, the problems are fundamentally too difficult for the model to solve. We acknowledge that improving the inherent reasoning capabilities for these highly complex edge cases is a compelling direction for future research
>
> ***Q1: Generalization on Other Languages and Threshold Sensitivity***
>
> τ does not require manual recalibration and the DID density thresholds are not highly sensitive to languages other than Python. We detail the justifications below:
>
> (1) As detailed in Algorithm 1, τ is determined dynamically via the DetectValueSpike function. We identify this threshold by locating the most significant value gap (spike) in the ranked perplexity increment distribution during perturbation analysis. This data-driven approach naturally adapts to varying linguistic densities.
>
> (2) To directly address the sensitivity of the Distillation Information Density (DID) bounds to languages other than Python, we conduct an additional evaluation on C++. We train Llama3.2-3B using 1,846 C++ data sourced from MultiPL-E and MBXP across three different DID intervals, and test it on the HumanEval-X C++ dataset.
>
> By fitting the empirical performance data on the C++ dataset to our theoretical DID model (Formula 7), the derived optimal density peak remains highly stable at approximately **4.9%**, which remains within our original threshold range. This stability occurs because DID is derived from the information density of the plan and the LLM's statistical attention distribution over it, making the threshold naturally insensitive to the target programming language's syntactic density. We will incorporate these empirical results into the revised version.

---

> > ### Author Rebuttal · Reviewer_yLPD · 2026-04-02
> >
> > I thank the authors for the response. I am keeping my (positive) score.

---

> > > ### Author Response · Authors · 2026-04-02
> > >
> > > Thank you for your careful review and valuable suggestions. Your comments have helped us improve the paper by clarifying the computational overhead, strengthening the discussion on plan quality dependency, and adding more solid evidence regarding cross-language generalization and threshold robustness. We sincerely appreciate your constructive feedback, which has made the revised paper more rigorous, better supported empirically, and clearer to readers.

---

### Official Review · Reviewer_84rV · 2026-03-13

**Soundness:** 3
**Presentation:** 3
**Significance:** 3
**Originality:** 3
**Overall Recommendation:** 4
**Confidence:** 4

**Summary:**

This paper proposes PADA, a training-based framework to address Attention Allocation Imbalance in the Plan-then-Code paradigm for small-scale LLMs (<8B parameters). The authors identify two failure modes — Attention Dispersion (attention spread across irrelevant tokens) and Attention Drift (attention migrating away from the plan as generation progresses) — and address them through three components: (1) perturbation-verified key token selection using perplexity increment (ΔPPL) as an importance metric, guided by a Distillation Information Density (DID) framework to determine the optimal token density; (2) a Maximum-DID binary attention target matrix constructed from teacher–student consensus and disagreement token sets; and (3) dynamic attention alignment training with a progress-aware sliding window (width 3) and difficulty-aware gating. Experiments are conducted across three student models, two teacher models, and seven benchmarks, with PADA achieving up to 16.7% Pass@1 improvement over baselines.
The core problem is well-motivated and the perturbation-based token selection idea has some novelty.

**Compliance With Llm Reviewing Policy:**

Affirmed.

**Key Questions For Authors:**

How to "feed plans as prompts into different models" in Section 4.2?

**Limitations:**

yes

**Strengths And Weaknesses:**

## Strengths

1. Well-motivated problem with empirical grounding.

The two attention failure modes (Dispersion and Drift) are real phenomena in long-context generation. Appendix A provides informative visualizations: correct generations show a mean plan-region attention of 0.015 vs. 0.013 for incorrect ones, and Appendix A.2 statistically validates the sliding window width choice across 200 samples (Figure 11).

2. Large performance gains on weak models.

On Llama3.2-3B-instruct, PADA achieves a relative improvement of 40.8%, which is practically significant. The result suggests PADA is especially useful for instruction-following-limited models.

3. Perturbation-verified token selection is a meaningful contribution.

Using ΔPPL to distinguish genuinely critical tokens from high-attention but semantically empty ones (e.g., punctuation, whitespace) is a reasonable approach that addresses a real limitation of naive top-k attention selection. The DID optimality analysis in Appendix C (Figure 17), which verifies a unique maximum via numerical differentiation, adds mathematical rigor to the density selection procedure.

4. Causal validation rules out overfitting.

Appendix D uses an independent causal analysis framework (adapted from LeaF) to verify that the improved attention allocation corresponds to semantically causal tokens, not just training target overfitting. This strengthens the qualitative claims in RQ4.

##Weaknesses

1 Missing Self-Anchor Baseline

Self-Anchor (Zhang et al., 2025) is the most directly competing method and is explicitly cited in the Related Work section. Both Self-Anchor and PADA decompose reasoning trajectories into structured sub-steps and train the model to align its attention to each step during code generation. The key claimed differences of PADA — perturbation-verified token selection and DID-guided density control — are exactly what would need to be tested against Self-Anchor to establish PADA's incremental contribution.

The absence of Self-Anchor from the main experimental table makes it impossible to assess whether PADA's improvements come from the proposed mechanisms or simply from the general approach of step-wise attention alignment that both methods share. This is the most significant weakness of the paper.

2 Unfair Baseline Configuration for LPW

The main table presents LPW with only a single iteration (LPW@1), but iterative refinement via execution feedback is not an optional add-on to LPW — it is the core mechanism that defines the method. Restricting LPW to one iteration is equivalent to evaluating it without its primary contribution, producing a result that is not representative of the method as designed.
The authors justify this choice as a means of ensuring fairness, implying that one generation pass from PADA should be compared against one iteration of LPW. This reasoning is flawed. PADA and LPW operate on fundamentally different axes: PADA is a trained model that internalizes plan-following capability through offline distillation and requires only a single inference pass by design; LPW is an inference-time iterative refinement system whose performance scales with the number of feedback rounds. These are not commensurable units. Constraining LPW to one iteration does not "control" for anything meaningful — it simply removes the method's defining capability to make the comparison appear more favorable to PADA.

Appendix B.3 (Table 7) reveals the consequence of this choice: LPW@12 reaches 97.6% on HumanEval, surpassing PADA's 95.7%, and achieves 40.7% on APPS-C compared to PADA's 46.7%. The appropriate framing of PADA's advantage is therefore not that it outperforms LPW in absolute accuracy across the board, but that it achieves competitive or superior accuracy at a fraction of the token cost — approximately 3.6× fewer tokens on APPS and over 10× fewer on HumanEval. This is a genuinely strong and defensible claim. The decision to present LPW@1 in the main table instead of making this efficiency argument directly obscures what is actually PADA's most honest and compelling contribution.

3 Unspecified RQ2 inference procedure.

The paper describes RQ2 only as "feeding these plans as prompts into different models" (Section 4.2), with no further detail. PADA-Coder's standard inference pipeline involves multi-candidate plan generation, plan verification, and best-plan selection before code generation. It is unclear whether this pipeline is bypassed in RQ2 when an external plan is provided, or whether the model still internally generates and verifies its own candidate plans alongside the provided reference. If PADA-Coder's full inference pipeline is active in RQ2, then the external Claude plan is not truly "fixed" as a controlled input — the model may be partially ignoring it in favor of self-generated candidates, making the experiment inconsistent with its stated design.

The authors' choice of claude-sonnet-4.5 as the plan generator for RQ2 is never justified. Two more natural alternatives exist.
The first alternative is Qwen3-32B, the same teacher model used in the main experiments.  The authors do not explain why this option was not taken. The second alternative is to use the best plan produced by PADA-Coder's own inference pipeline as the fixed reference plan. PADA's standard inference procedure already generates multiple candidate plans and selects the best one via verification. This self-generated best plan is exactly what PADA uses in end-to-end deployment, and is therefore the most ecologically valid input for isolating the code generation step. Fixing this plan and feeding it to all compared models would directly answer the question "given the plan that PADA actually produces, how well can each model execute it?"

4 Critical Dependency on Plan Quality (Hidden Deployment Cost)

The experiments in Appendix B.1 explicitly demonstrate that when provided with incorrect plans, PADA-Coder's performance (APPS: 30.8%) falls marginally below that of the base model (APPS: 32.9%). This exposes a fundamental limitation of the proposed method: the observed performance gains are entirely contingent on high-quality plan inputs. However, obtaining such high-quality plans itself requires a stronger LLM — ground-truth-guided plan construction during training and multi-candidate selection at inference time — introducing substantial deployment overhead that is significantly understated in the main paper. The authors do not adequately acknowledge this dependency in the primary experimental section, which risks overstating the practical utility of the approach.

---

> ### Author Rebuttal · Authors · 2026-03-31
>
> Thank you for recognizing the empirical motivation and the mathematical rigor of PADA. We provide the following responses regarding the weaknesses you pointed out.
>
> ***W1: Missing Baseline***
>
> Our initial omission of Self-Anchor was because it is closed-source and originally optimized for mathematical reasoning rather than code generation. Unlike math's sequential logic, code generation demands strict syntax, execution constraints, and long-range dependencies.
>
> However, inspired by your feedback, we reproduce the Self-Anchor mechanism for code generation benchmarks. To guarantee a strictly fair evaluation, we test Self-Anchor under the exact same settings as PADA. The comparative results are presented below:
>
> Table R1: Comparative performance of PADA and Self-Anchor
> |Method|HumanEval|MBPP|MBPP+|APPS-I|APPS-V|APPS-C|LCB|Average|
> |-|-|-|-|-|-|-|-|-|
> |Self-Anchor|93.8|85.9|73.5|74.8|55.4|36.6|61.3|68.5|
> |PADA-Coder|96.8|93.2|85.5|79.3|65.2|50.4|74.4|77.8|
>
> As shown in table R1, PADA consistently outperforms Self-Anchor. While Self-Anchor aligns attention to the entire planning step, it suffers from "Attention Dispersion" (as visually defined in Figure 1) across non-key tokens. We will incorporate the empirical results into the revised version.
>
> ***W2: Unfair Baseline***
>
> We sincerely thank the reviewer for pointing out this baseline configuration. We agree that LPW@12 represents the optimal capability of the LPW framework. In fact, we have already provided a comparison between PADA and LPW@12 across 7 datasets in Appendix B.3 (Table 7). As detailed in Table 7, PADA achieves a higher average accuracy and outperforms LPW@12 on 5 out of the 7 datasets, particularly on complex tasks like APPS-C (46.7% vs 40.7%). However, we acknowledge that on 2 simpler datasets (HumanEval and MBPP+), LPW@12 does achieve higher accuracy (e.g., 95.7% vs 97.6% on HumanEval) through its iterative execution.
>
> In the revised version, we will move this comparison to the primary experimental section, replacing LPW@1 with LPW@12. We will revise our claims to reflect that rather than achieving absolute accuracy superiority across all tasks, PADA achieves higher overall average performance, while maintaining an advantage in inference efficiency (as shown in Figure 13).
>
> ***W3 & Q1: RQ2 Inference Procedure and Plan Generator***
>
> **1.Inference Procedure in RQ2:** When testing with reference plans, PADA-Coder's internal multi-candidate generation pipeline is bypassed. The model strictly uses the provided plan as a fixed prompt and proceeds to code generation after a single validation with Public Test Cases.
>
> **2.Reason for Choosing Claude-Sonnet-4.5:** As shown in Table R2, Claude-Sonnet-4.5 can achieve near-oracle correctness in our verification pipeline, whereas small-scale LLMs struggle with a small subset of the most difficult problems. Using Claude ensures that our evaluation of plan-following (code generation) capability is not bottlenecked by the model's planning limits.
>
> Table R2: Plan Generation Accuracy Comparison
> | Model|APPS|HumanEval|MBPP+|
> |-|-|-|-|
> |Claude-Sonnet-4.5|99.7%|100%|100%|
> |Qwen3-32B|95.2%|100%|97.8%|
> |PADA-Coder(Qwen3-4B)|92.3%|100%|96.7%|
>
> **3.The Bias of Using Self-Generated Plans:** Due to capacity limits, PADA-Coder (Qwen3-4B) fails to generate correct plans for 58 highly difficult APPS samples. To prove this is a capacity bottleneck rather than a flaw in our method, we test these 58 samples using other methods on Qwen3-4B: LPW@12 solved only 3 (5.2%), and LeaF solved only 2 (3.4%). If we strictly use self-generated plans, these hardest problems are excluded from the evaluation. Despite this bias, we conduct the exact experiment you suggested: fixing the correct plans generated by PADA-Coder (Qwen3-4B) and feeding them to all models.
>
> Table R3: Code Generation Performance based on PADA-Coder's Self-Generated Correct Plans
> |Model|APPS|HumanEval|MBPP+|Average|
> |-|-|-|-|-|
> |DeepSeek-v3.2|67.6%|95.7%|88.6%|84.0%|
> |Gemini-2.5-Pro|72.1%|99.3%|91.7%|87.7%|
> |GPT-5.2|70.6%|98.7%|90.4%|86.6%|
> |PADA-Coder|81.5%|98.2%|93.5%|91.1%|
>
> As shown in table R3, Since the 58 hardest problems are filtered out, all models' accuracies improve, with PADA-Coder gaining the most. This demonstrates why using Claude as an Oracle is a fairer benchmark.
>
> ***W4: Dependency on Plan Quality***
>
> We politely disagree that achieving high-quality plans requires substantial overhead or massive LLMs, although we acknowledge that incorrect plans can cause a slight degradation in accuracy.
>
> First, we have analyzed the impact of incorrect plans in Appendix B.1 (Table 6). This analysis demonstrates that our model maintains robustness. Furthermore, as demonstrated in Table R2, the 4B-parameter PADA-Coder already achieves a high plan generation accuracy. Therefore, a single deployment of the small-scale PADA-Coder is sufficient for most cases in our experiments to independently generate accurate plans and execute them for the majority of programming tasks.

---

> > ### Author Rebuttal · Reviewer_84rV · 2026-04-02
> >
> > The rebuttal resolved my concern.

---

> > > ### Author Response · Authors · 2026-04-02
> > >
> > > Thank you for your careful review and valuable suggestions. Your comments have helped us improve the paper by incorporating the highly relevant Self-Anchor baseline, properly reframing the performance-efficiency trade-off against the complete LPW@12 framework, clarifying our inference pipeline, and transparently discussing the impact of plan quality on deployment costs. We sincerely appreciate your supportive assessment, and we believe these revisions have made the paper clearer, more rigorous, and easier to follow.

---

### Decision · Program_Chairs · 2026-04-30

**Decision:**

Accept (regular)

**Comment:**

This paper proposes PADA-Coder, a training-based method for improving plan-following code generation in small LLMs. It has following strengths:

* Multiple reviewers agreed that this is a well-motivated problem.
* Perturbation-based token selection using ΔPPL to filter out high-attention but semantically empty tokens was viewed as a novel contribution.
* The empirical results were consistently strong across baselines , with especially large gains on weaker models.

The main concerns were the missing Self-Anchor baseline, the unfair LPW comparison in the main table, the under-discussed dependency on plan quality and deployment overhead, the fact that the evidence was stronger for moderate composite prompts than for the broadest long-context framing, and the lack of clarity around key implementation details such as plan-code alignment and the RQ2 inference setup.
After rebuttal, all reviewers indicated that concerns had been addressed.

Reviewer AnDX, in fit justification, have two concerns about long composite prompts and difficulty-aware gating mechanism, but AnDX no longer saw it as outweighing the paper’s practical contribution.
Overall, I believe this paper should be accepted.